

# A comparison of the performance of depth-integrated ice-dynamics solvers

Alexander Robinson[1,2,3,4], Daniel Goldberg[5], and William H. Lipscomb[4]

[1]Complutense University of Madrid, Madrid, Spain
[2]Geosciences Institute CSIC-UCM, Madrid, Spain
[3]Potsdam Institute for Climate Impact Research, Potsdam, Germany
[4]Climate and Global Dynamics Laboratory, National Center for Atmospheric Research, Boulder, CO 80305, USA
[5]School of GeoSciences, University of Edinburgh, Edinburgh, UK

**Correspondence:** Alexander Robinson (robinson@ucm.es)

**Abstract.** In the last decade, the number of ice sheet models has increased substantially, in line with the growth of the glaciological community. These models use solvers based on different approximations of ice dynamics. In particular, several depth-integrated dynamics solvers have emerged as fast solvers capable of resolving the relevant physics of ice sheets at the continental scale. However, the numerical stability of these schemes has not been studied systematically to evaluate their effectiveness

in practice. Here we focus on three such solvers, the so-called Hybrid, L1L2-SIA and DIVA solvers, as well as the well-known SIA and SSA solvers as boundary cases. We investigate the numerical stability of these solvers as a function of grid resolution and the state of the ice sheet. Under simplified conditions with constant viscosity, the maximum stable timestep of the Hybrid solver, like the SIA solver, has a quadratic dependence on grid resolution. In contrast, the DIVA solver has a maximum timestep that is independent of resolution as the grid become increasingly refined, like the SSA solver. Analysis indicates that

the L1L2-SIA solver should behave similarly, but in practice, the complexity of its implementation can make it difficult to maintain stability. In realistic simulations of the Greenland ice sheet with a non-linear rheology, the DIVA and SSA solvers maintain superior numerical stability, while the SIA, Hybrid and L1L2-SIA solvers show markedly poorer performance. At a grid resolution of $\Delta x = 4\,\text{km}$, the DIVA solver runs approximately 15 times faster than the Hybrid and L1L2-SIA solvers as a result of a larger stable timestep. Our analysis shows that as resolution increases, the ice-dynamics solver can act as a

bottleneck to model performance. The DIVA solver emerges as a clear outlier in terms of both model performance and its representation of the ice-flow physics itself.

## 1 Introduction

Modeling ice sheets at the continental scale requires compromise. The Greenland and Antarctic ice sheets span thousands of kilometers, and interact with the rest of the Earth system on timescales of up to thousands of years. Solving the full Stokes

stress balance to calculate the ice dynamics over such a large domain for a long timescale is still computationally infeasible. Therefore, it is important to reduce the complexity of simulations of ice dynamics.





Several factors reduce the need to simulate the whole system using the full Stokes equations. First, ice sheets typically have low aspect ratios, growing only a few kilometers thick near the domes, but extending thousands of kilometers horizontally (i.e., ice sheets are shallow). This condition implies that over most length scales of interest, several terms in the stress balance can be ignored. Furthermore, some boundary conditions, particularly the basal friction at the bed, are poorly known, which means

that the added accuracy in simulating velocities with a full Stokes approach may not translate into more robust estimates of past or future ice-sheet evolution.

Many approximations to the full Stokes stress balance exist, each with trade-offs. The simplest and most widely used shallow approximation is the shallow ice approximation (SIA) (Hutter, 1983; Morland, 1984). The SIA is valid for slow-flowing grounded ice, frozen to the bedrock, which equates to shear flow induced by gravitational driving stress balanced by basal

drag. It is a local solution, in that the velocity diagnosed at a given location is fully determined by the local driving stress. A complementary, but non-local, approximation is the shallow-shelf approximation (SSA) (Morland, 1987; MacAyeal, 1989), which represents fast-flowing ice that is either floating or sliding rapidly over the bed, resulting in plug flow (i.e., with a constant vertical profile of horizontal velocity).

An ad-hoc approach to gaining the benefits of computational speed and validity in multiple flow regimes is to combine

the SIA and SSA by summing their contributions to obtain the "hybrid" horizontal velocity field (hereafter called the Hybrid solver). This approach was proposed by Bueler (2009), who used a weighting function to transition between the two regimes. Later, Winkelmann et al. (2011) recognized that, in light of the approximations involved and the uncertainty in basal friction, it is more straightforward simply to sum the contributions. The fundamental assumption behind this approach is that the SSA represents a sliding regime, in which the depth-averaged velocity is equal to the basal velocity, while the SIA represents a

frozen regime, in which the basal velocity is zero. In the former, it can be expected that SIA velocities will be negligible and in the latter, SSA velocities will be zero. Only in the narrow transition between the two will the velocity solutions be mixed, and the Hybrid approach should provide a smooth transition. The Hybrid solver is used by several models today (e.g., Winkelmann et al., 2011; Pattyn, 2017; Quiquet et al., 2018; Robinson et al., 2020). In fact, roughly half the models participating in ISMIP6-Greenland and ISMIP6-Antarctica used a Hybrid solver (Goelzer et al., 2018; Seroussi et al., 2019;

Goelzer et al., 2020; Seroussi et al., 2020).

Others have rigorously derived approximations based on variational principles that intrinsically account for both shear and stretching (Hindmarsh, 2004; Schoof and Hindmarsh, 2010; Goldberg, 2011). The depth-integrated approximations are especially appealing, since they are comparable in speed per timestep to the SSA and Hybrid solvers. The depth-integrated solver derived by Schoof and Hindmarsh (2010) and formalized by Perego et al. (2012) is part of the L1L2 family of solvers, fol-

lowing the terminology of Hindmarsh (2004). We refer to it here as the L1L2-SIA solver. The stress balance is solved at one layer in the ice sheet – in this case, at the bed – while the effective viscosity accounts for stress arising from both shearing and stretching. Thus the solver enables a 2D solution of the system of partial differential equations (PDEs) and fast performance compared to 3D solvers. This solver naturally incorporates both shearing and sliding regimes while approximating the shear stress components via the SIA, which facilitates its numerical solution. The L1L2-SIA solver is used by the BISICLES model

(Cornford et al., 2013).



The so-called depth-integrated viscosity approximation (DIVA) derived by Goldberg (2011) is also part of the L1L2 family of solvers. Here, the stress balance is formulated in terms of the depth-averaged velocity, but like the L1L2-SIA solver, the viscosity accounts for both shear and stretching. The shear stress, however, is not approximated, and the longitudinal and lateral stresses are treated as depth-independent, which facilitates a 2D solution of the system of PDEs. The viscosity is vertically averaged, and basal friction is cast in terms of the depth-averaged velocity. This solver has been used by continental-scale models to investigate dynamics and interactions with climate for the Greenland and Antarctic ice sheets (Arthern et al., 2015; Arthern and Williams, 2017; Lipscomb et al., 2019, 2021).

All the approximations described above have been used effectively in a variety of glaciological contexts, but there has been no rigorous, comparative study of their numerical performance. Until recently, the primary concern at the regional/continental scale has been to represent the physics of ice flow, i.e., ensuring that both sliding and shearing regimes are well represented – a motivation that contributed to the development of the Hybrid, L1L2-SIA, and DIVA solvers. Today, however, the community is focused on running such simulations at high resolutions (i.e., $\Delta x < 5\,\mathrm{km}$), as the tools and computational resources become available.

Given the importance of computational efficiency in modelling glacial dynamics at the continental scale, the goal of this paper is to investigate the numerical performance of the solvers described above, when coupled to mass-conservation timestepping. We focus on the latter three solvers, which account for the dominant stress terms over most of an ice sheet, while the SIA and SSA solvers are useful boundary cases. Below, we first outline the numerical approach used by each of these solvers. Next, we derive stable timestep limits of ice-thickness advection in an analytical test case for the DIVA, Hybrid and L1L2-SIA solvers, and we investigate the underlying factors that can affect the maximum stable timestep given each solver. The analytical results are validated in idealized numerical tests. We then compare the performance of all five solvers in terms of stable timestep size and model computational speed in a realistic test case of quasi steady-state simulations of the Greenland ice sheet. This is followed by a discussion of the results and conclusions.

## 2 Ice dynamics solvers

We describe the assumptions and equations behind the five solvers considered here, namely: the SIA, SSA, DIVA, Hybrid and L1L2-SIA solvers. These approximations can be obtained by considering the various terms in the first-order Blatter-Pattyn (BP) approximation (Blatter, 1995; Pattyn, 2003). To give context to the depth-integrated solvers, we first write the basic equations of the 3D BP stress-balance approximation:

$$
\frac{\partial}{\partial x}\left[2\mu\left(2\frac{\partial u}{\partial x}+\frac{\partial v}{\partial y}\right)\right] + \frac{\partial}{\partial y}\left[\mu\left(\frac{\partial u}{\partial y}+\frac{\partial v}{\partial x}\right)\right] + \frac{\partial}{\partial z}\left(\mu\frac{\partial u}{\partial z}\right) = \rho_i g\frac{\partial s}{\partial x},
$$
$$
\frac{\partial}{\partial y}\left[2\mu\left(2\frac{\partial v}{\partial y}+\frac{\partial u}{\partial x}\right)\right] + \frac{\partial}{\partial x}\left[\mu\left(\frac{\partial u}{\partial y}+\frac{\partial v}{\partial x}\right)\right] + \frac{\partial}{\partial z}\left(\mu\frac{\partial v}{\partial z}\right) = \rho_i g\frac{\partial s}{\partial y},
$$
$$
\tag{1}
$$

where $u$ and $v$ are the components of horizontal velocity, $\mu$ is the effective viscosity, $\rho_i$ is the density of ice (assumed constant), $g$ is gravitational acceleration, $s$ is the surface elevation, and $x, y, z$ are 3D Cartesian coordinates. In each equation the three





terms on the left-hand side (LHS) describe gradients of longitudinal stress, lateral shear stress, and vertical shear stress, respectively, and the right-hand side (RHS) is the gravitational driving force. The longitudinal and lateral shear stresses together are also referred to as membrane stresses (Hindmarsh, 2006).

The effective viscosity $\mu$ is defined as

$$\mu \equiv \frac{1}{2}A^{-\frac{1}{n}}\dot{\varepsilon}_e^{\frac{1-n}{n}}, \tag{2}$$

where $A$ is the temperature-dependent rate factor in Glen's flow law (Glen, 1955), $n$ is the Glen exponent (many models set $n = 3$), and $\dot{\varepsilon}_e$ is the effective strain rate, given in the BP approximation by

$$\dot{\varepsilon}_e^2 = u_x^2 + v_y^2 + u_x v_y + \frac{1}{4}(u_y + v_x)^2 + \frac{1}{4}u_z^2 + \frac{1}{4}v_z^2. \tag{3}$$

Here, $u_x$ denotes the partial derivative $\partial u/\partial x$, and similarly for other derivatives.

## 2.1 SIA solver

The shallow-ice approximation (SIA) solver is a zero-order approximation to ice flow that assumes a balance between the basal drag and the gravitational driving stress (Greve and Blatter, 2009). This leads to purely shear-stress-driven flow within the ice column. In other words, the membrane stress gradients are ignored, which leads to a stress balance equation that can be solved locally. This is a typical flow regime for grounded ice that is not sliding.

By setting membrane stress gradients to zero in Eq. (1), we obtain the SIA stress balance equations:

$$\frac{\partial}{\partial z}\left(\mu\frac{\partial u}{\partial z}\right) = \rho_i g\frac{\partial s}{\partial x},$$
$$\frac{\partial}{\partial z}\left(\mu\frac{\partial v}{\partial z}\right) = \rho_i g\frac{\partial s}{\partial y}, \tag{4}$$

where the effective viscosity $\mu$ in Eq. (2) is obtained using an effective strain rate that includes only the shear terms:

$$\dot{\varepsilon}_e^2 = \frac{1}{4}\left(u_z^2 + v_z^2\right). \tag{5}$$

By assuming a low aspect ratio and a free-stress surface boundary condition, these equations can readily be integrated from the bed $b$ to the surface $s$ to provide a 2D solution for depth-average ice flow:

$$\bar{u} = -\frac{\rho_i g H^2}{3\bar{\mu}}\frac{\partial s}{\partial x},$$
$$\bar{v} = -\frac{\rho_i g H^2}{3\bar{\mu}}\frac{\partial s}{\partial y}, \tag{6}$$

where $\bar{u}$ and $\bar{v}$ are 2D vertically averaged velocity components, and $\bar{\mu}$ is the vertically averaged effective viscosity:

$$\bar{u} = \frac{1}{H}\int_b^s u(z)dz, \quad \bar{v} = \frac{1}{H}\int_b^s v(z)dz, \quad \bar{\mu} = \frac{1}{H}\int_b^s \mu(z)dz. \tag{7}$$



## 2.2 SSA solver

The shallow-shelf approximation (SSA) solver is complementary to the SIA solver in that the membrane stress gradients are retained, while the vertical shear stress gradients are ignored. This is a typical flow regime for floating ice or rapidly sliding ice streams, where the driving stress is low and the ice column has a uniform velocity profile (i.e., plug flow). By dropping the vertical shear-stress gradient terms in Eq. (1) and integrating vertically while assuming a vertically uniform velocity, we obtain the 2D SSA stress balance equations:

$$\frac{\partial}{\partial x}\left[2\bar{\mu}H\left(2\frac{\partial u}{\partial x}+\frac{\partial v}{\partial y}\right)\right]+\frac{\partial}{\partial y}\left[\bar{\mu}H\left(\frac{\partial u}{\partial y}+\frac{\partial v}{\partial x}\right)\right]-\tau_{b,x}=\rho_i gH\frac{\partial s}{\partial x},$$
$$\frac{\partial}{\partial y}\left[2\bar{\mu}H\left(2\frac{\partial v}{\partial y}+\frac{\partial u}{\partial x}\right)\right]+\frac{\partial}{\partial x}\left[\bar{\mu}H\left(\frac{\partial u}{\partial y}+\frac{\partial v}{\partial x}\right)\right]-\tau_{b,y}=\rho_i gH\frac{\partial s}{\partial y}. \tag{8}$$

where $\tau_{b,i}$ indicates the basal stress in the direction $i$. We assume a basal friction law of the form

$$\boldsymbol{\tau_b}=\beta\boldsymbol{u_b}, \tag{9}$$

where $\boldsymbol{\tau_b}$ is the basal shear stress, $\boldsymbol{u_b}=(u_b,v_b)$ is the basal velocity, and $\beta$ is a non-negative friction coefficient that can depend on $\boldsymbol{u_b}$. The effective viscosity is calculated following Eq. (2) for the BP solver, using a vertically averaged rate factor and computing the effective strain rate in 2D without the shear terms:

$$\dot{\varepsilon}_e^2=u_x^2+v_y^2+u_xv_y+\frac{1}{4}(u_y+v_x)^2. \tag{10}$$

## 2.3 DIVA solver

Goldberg (2011) derived a higher-order stress approximation that is similar in accuracy to the first-order BP stress balance, but like other depth-integrated solvers is computationally much cheaper. Since the stress-balance equations use a depth-integrated effective viscosity in place of a vertically varying viscosity, we refer to this scheme as a depth-integrated-viscosity approximation, or DIVA.

In the DIVA solver, the horizontal velocity gradients and effective viscosity in the membrane stress terms are replaced by vertical averages $\bar{u}$, $\bar{v}$, and $\bar{\mu}$ (Eq. (7)). The 3D effective viscosity is given by Eq. (2), but with the DIVA effective strain rate:

$$\dot{\varepsilon}_e^2=\bar{u}_x^2+\bar{v}_y^2+\bar{u}_x\bar{v}_y+\frac{1}{4}(\bar{u}_y+\bar{v}_x)^2+\frac{1}{4}u_z^2+\frac{1}{4}v_z^2, \tag{11}$$

The vertically integrated stress balance in DIVA can be written as

$$\frac{\partial}{\partial x}\left[2\bar{\mu}H\left(2\frac{\partial\bar{u}}{\partial x}+\frac{\partial\bar{v}}{\partial y}\right)\right]+\frac{\partial}{\partial y}\left[\bar{\mu}H\left(\frac{\partial\bar{u}}{\partial y}+\frac{\partial\bar{v}}{\partial x}\right)\right]-\tau_{b,x}=\rho_i gH\frac{\partial s}{\partial x},$$
$$\frac{\partial}{\partial y}\left[2\bar{\mu}H\left(2\frac{\partial\bar{v}}{\partial y}+\frac{\partial\bar{u}}{\partial x}\right)\right]+\frac{\partial}{\partial x}\left[\bar{\mu}H\left(\frac{\partial\bar{u}}{\partial y}+\frac{\partial\bar{v}}{\partial x}\right)\right]-\tau_{b,y}=\rho_i gH\frac{\partial s}{\partial y}, \tag{12}$$

where the boundary conditions at $b$ and $s$ have been used to evaluate the vertical stress terms, and the basal stress is defined as for the SSA solver. Equation (12) has the same form as the SSA stress balance, but the dependence of basal stress and viscosity on velocity is more complex.





To solve Eq. (12) for the mean horizontal velocity components, the basal stress terms must be written in terms of $\bar{u}$ and $\bar{v}$. The following discussion refers to the $u$ component; the $v$ component is analogous. Goldberg (2011) showed that the vertical profile of $u$ is related to $u_b$ by

$$u(z) = u_b + \frac{\beta u_b}{H} \int\limits_b^z \frac{(s - z')}{\mu(z')} dz'. \tag{13}$$

Following Arthern et al. (2015), we define some generalized integrals $\mathcal{F}_m$ for clarity:

$$\mathcal{F}_m \equiv \int\limits_b^s \frac{1}{\mu} \left( \frac{s - z}{H} \right)^m dz. \tag{14}$$

The surface velocity is thus related to the bed velocity as

$$u_s = u_b \left( 1 + \beta \mathcal{F}_1 \right). \tag{15}$$

Integrating $u(z)$ from the bed to the surface gives the depth-averaged mean velocity $\bar{u}$:

$$\bar{u} = u_b \left( 1 + \beta \mathcal{F}_2 \right), \tag{16}$$

which allows $\beta u_b$ in Eq. (9) to be replaced with $\beta_{\text{eff}} \bar{u}$, where

$$\beta_{\text{eff}} \equiv \frac{\beta}{1 + \beta \mathcal{F}_2}. \tag{17}$$

For a frozen bed ($u_b = 0$, with nonzero basal stress $\tau_{b,x}$), the $\beta u_b$ term on the RHS of Eq. (13) is replaced by $\tau_{b,x}$, leading to

$$\bar{u} = \tau_{b,x} \mathcal{F}_2. \tag{18}$$

Then the basal stress term $\beta u_b$ in Eq. (9) can be replaced by $\beta_{\text{eff}}^{\text{frz}} \bar{u}$, where

$$\beta_{\text{eff}}^{\text{frz}} \equiv \frac{1}{\mathcal{F}_2}. \tag{19}$$

The velocity is found in two steps. First, Eq. (12) is solved for the mean velocity, using Eqs. (9), (17), and (19) to write the basal stress terms in terms of $\bar{u}$ and $\bar{v}$. Then Eq. (13) is integrated vertically to find the 3D velocity.

## 2.4   Hybrid solver

The Hybrid solver, as defined here, follows the approach of Winkelmann et al. (2011). The horizontal velocity is defined as the sum of the depth-dependent internal shear velocity $(u_i, v_i)$ and the basal sliding velocity $(u_b, v_b)$:

$$u = u_i + u_b,$$
$$v = v_i + v_b. \tag{20}$$





The sliding velocity is calculated via the SSA and the internal shear velocity is calculated via the SIA, as defined above. This approach generally works well because over most regions of an ice sheet, either plug flow (sliding) or shear flow is dominant, so one approximation is sufficient to represent the flow, and the other term goes to zero (Winkelmann et al., 2011; Pollard and DeConto, 2012). Although this approach is used widely with success at reproducing observed ice flow (e.g. Winkelmann et al., 2011; Quiquet et al., 2018), it is less clear how well it resolves the transition between the two regimes.

## 2.5 L1L2-SIA solver

The L1L2-SIA solver as defined here follows from Schoof and Hindmarsh (2010) and Perego et al. (2012). Like the DIVA and Hybrid balances, it is a two-step approach that begins by solving a depth-averaged stress balance. The balance is similar in in form to the SSA (Eq. (8)):

$$
\frac{\partial}{\partial x}\left[2\bar{\mu}'H\left(2\frac{\partial u_b}{\partial x}+\frac{\partial v_b}{\partial y}\right)\right]+\frac{\partial}{\partial y}\left[\bar{\mu}'H\left(\frac{\partial u_b}{\partial y}+\frac{\partial v_b}{\partial x}\right)\right]-\tau_{b,x}=\rho_i gH\frac{\partial s}{\partial x},
$$
$$
\frac{\partial}{\partial y}\left[2\bar{\mu}'H\left(2\frac{\partial v_b}{\partial y}+\frac{\partial u_b}{\partial x}\right)\right]+\frac{\partial}{\partial x}\left[\bar{\mu}'H\left(\frac{\partial u_b}{\partial y}+\frac{\partial v_b}{\partial x}\right)\right]-\tau_{b,y}=\rho_i gH\frac{\partial s}{\partial y}.
$$
$$(21)$$

Aside from the fact that the equations are solved for basal, not depth-uniform, velocity, the balance differs from the SSA in the form of the effective viscosity. $\bar{\mu}'$ is a vertical average of $\mu'$, the L1L2 effective viscosity, which is given by

$$
\mu'=\frac{1}{2}A^{-1}\left(\tau_{b,e}^2+[\rho_i g(s-z)]^2\left[\left(\frac{\partial s}{\partial x}\right)^2+\left(\frac{\partial s}{\partial y}\right)^2\right]\right)^{\frac{1-n}{2}},
$$
$$(22)$$

where

$$
\tau_{b,e}^2=\tau_{b,xx}^2+\tau_{b,yy}^2+\tau_{b,xx}\tau_{b,yy}+\tau_{b,xy}^2,
$$
$$(23)$$

with $\tau_{b,ij}$, $i,j=\{x,y\}$ the longitudinal stress induced by the basal strain-rate tensor. $\tau_{b,e}$ depends implicitly on $\dot{\varepsilon}_{b,e}$, the effective horizontal (2D) strain rate induced by the basal strain-rate tensor (*cf* Eq. (10)):

$$
A\left(\tau_{b,e}^2+[\rho_i g(s-z)]^2\left[\left(\frac{\partial s}{\partial x}\right)^2+\left(\frac{\partial s}{\partial y}\right)^2\right]\right)^{\frac{n-1}{2}}\tau_{b,e}=\dot{\varepsilon}_{b,e}.
$$
$$(24)$$

For $n=3$, taking $\dot{\varepsilon}_{b,e}$ as a function of $u_b$ and $v_b$ from the previous iteration, Eq. (24) is a cubic equation for $\tau_{b,e}$ and can be solved analytically. Given $\bar{\mu}'$ from Eqs. (22) and (24), we can then solve Eq. (21) for $u_b$ and $v_b$. Note that the "trial" viscosity $\mu'$ makes the approximation that vertical shear stress is equal to the shallow-ice shear stress, $-\rho_i gH\nabla s$. In the second step, three-dimensional vertical shear stress is diagnosed from the solution:

$$
\tau_{xz}=\frac{\partial}{\partial x}\left[2\mu''(z)\left(2\frac{\partial u_b}{\partial x}+\frac{\partial v_b}{\partial y}\right)\right]+\frac{\partial}{\partial y}\left[\mu''(z)\left(\frac{\partial u_b}{\partial y}+\frac{\partial v_b}{\partial x}\right)\right]-\rho_i g(s-z)\frac{\partial s}{\partial x},
$$
$$(25)$$

where

$$
\mu''(z)=\int_z^s \mu' dz.
$$
$$(26)$$





$\tau_{yz}$ is diagnosed with a similar expression. Finally, we can define an updated viscosity:

$$\mu = \frac{1}{2} A^{-1} \left( \tau_{b,e}^2 + \tau_{xz}^2 + \tau_{yz}^2 \right)^{\frac{1-n}{2}}. \tag{27}$$

Horizontal velocities are then given by

$$\{u(z), v(z)\} = \{u_b, v_b\} + 2 \int_b^z \mu^{-1}\{\tau_{xz}, \tau_{yz}\}dz. \tag{28}$$

An important point regarding L1L2-SIA is the difference between $\mu'$ and $\mu$. The former is defined using the SIA shear stress as a proxy for the true vertical shear, while $\mu$ is formulated with an updated vertical shear based on the solution to the stress balance. This is in contrast to DIVA, where the viscosity is consistent between the two steps.

## 3 Analytical stability limits

In this section, we assess the numerical stability of various schemes to solve the coupled stress-balance and continuity equa-
tions. Our analysis is closely related to von Neumann stability analysis (Isaacson and Keller, 2012), but the non-local nature of the equations requires us to consider global solutions to the stress balance. For this analysis, we impose several simplifications:

- We reduce the problem to one horizontal dimension by setting $\bar{v} = 0$ and ignoring the $y$ derivatives.

- We assume that the effective viscosity $\mu$ is spatially uniform. This is equivalent to assuming that the ice is isothermal with $n = 1$ in Eq. (2) and the rate factor is constant, set to $A = (2\mu)^{-1}$. This also implies that $\mathcal{F}_2 = H/(3\mu)$ following
Eq. (14).

- We consider perturbations to an idealised geometry: an $x$-periodic domain with uniform thickness, $H_0$, and surface slope, $ds/dx = -\alpha$, where $\alpha > 0$.

This problem is described by Dukowicz (2012), who derived exact solutions for the velocity profile as a function of $\alpha$ and the non-dimensional constant $\eta = \beta H/\mu$.
We specifically analyze the DIVA, Hybrid and L1L2-SIA solvers. We begin with DIVA, which leads to the simplest expressions for stability in this framework; results for the Hybrid and L1L2-SIA balances follow. The SSA analysis arises naturally, since the SSA equations have the same form as the DIVA solver. The numerical stability of an SIA solver under the above assumptions has been done previously (Hindmarsh, 2001; Cheng et al., 2017), showing that the maximum stable timestep under the simplifications above is proportional to the square of the grid resolution:

$$\Delta t < \left( \frac{3\mu}{2\rho_i g H^3} \right) \Delta x^2 \tag{29}$$

We caution that the results depend on details of the numerical schemes and may not apply to all situations, such as when the rheology is nonlinear. Our aim is not to consider all possible schemes, or to produce a numerical scheme of optimal stability





as in Cheng et al. (2017), but rather to examine stability properties when applying a representative finite-difference scheme to different stress-balance equations. Our analytical results provide context for the empirical timestep limits found in more realistic settings in Section 4, and give a theoretical basis for those results. Readers interested primarily in the performance of various depth-integrated stress balances in continental-scale ice sheet models may wish to skip this section.

## 5    3.1    DIVA solver stability

We start with the $x$ component of the DIVA stress balance, Eq. (12). By assuming a constant slope $\alpha$ and spatially uniform effective viscosity $\mu$ and by setting $\bar{v} = 0$, this equation can be written as a second-order ordinary differential equation (ODE) for $\bar{u}$:

$$4\mu \frac{d}{dx}\left(H\frac{d\bar{u}}{dx}\right) - \left(\frac{3\beta}{3+\eta}\right)\bar{u} = \rho_i g H \frac{ds}{dx}. \tag{30}$$

The first term on the LHS is the longitudinal stress, and the RHS is the driving stress. In the second term on the LHS, the quantity in parentheses is $\beta_{\text{eff}}$, the effective basal friction coefficient. This coefficient includes a non-dimensional parameter, $\eta \equiv \beta H/\mu$, that relates basal friction to ice viscosity. If $\eta$ is small, we have $\beta_{\text{eff}} \approx \beta$, and the flow is dominated by basal sliding. If $\eta$ is large, then $\beta_{\text{eff}} \approx 3\mu/H$, and most of the flow is internal shearing. For the remainder of the analysis, $\beta$ is considered uniform.

To analyze the stability of a simple numerical scheme for the DIVA equations, we consider a reference state with uniform thickness and surface slope as described above. This reference state has uniform $\bar{u}$ and hence zero longitudinal stress, with depth-mean velocity

$$u_0 = \rho_i g H_0 \alpha \left(\frac{3+\eta}{3\beta}\right), \tag{31}$$

The thickness evolution equation is written as

$$\frac{\partial H}{\partial t} = -\frac{\partial}{\partial x}(\bar{u}H). \tag{32}$$

We will consider the evolution of thickness under an initial sinusoidal perturbation:

$$\delta H(t=0) = \varepsilon_h e^{ikx}. \tag{33}$$

Following Cheng et al. (2017), we approximate the DIVA momentum–mass balance, Eqs. (30) and (32), to first order in $\varepsilon_h$:

$$4\mu H_0 \frac{d^2\delta\bar{u}}{dx^2} - \frac{3\beta}{3+\eta}\delta\bar{u} = \rho_i g H_0 \frac{\partial \delta H}{\partial x} - \rho_i g \alpha \delta H, \tag{34}$$

$$\frac{\partial \delta H}{\partial t} = -u_0 \frac{\partial \delta H}{\partial x} - H_0 \frac{d\delta\bar{u}}{dx}, \tag{35}$$

where $\delta\bar{u}(x)$ is a perturbation in the mean velocity.



Next, we define a simple finite-difference scheme to solve this system of equations. While the theoretical domain has infinite extent, the computational domain must be finite, and we consider a periodic domain of length $L$. The discretised perturbation thickness, $h_n$, is located at points $x = n\Delta x$, for $n = 0,..,N$ (where $\Delta x = L/N$), and the discretised perturbation velocity, $u_n$, lies at points $(n+\frac{1}{2})\Delta x$, for $n = 0,..,N-1$. We assume that $L$ is much larger than the parameter $L_m$, defined as

$$5 \quad L_m = 2H_0\sqrt{\frac{3+\eta}{3\eta}}. \tag{36}$$

As shown in Appendix A1, $L_m$ is equivalent to a *membrane stress length scale*, over which longitudinal stresses are important (Hindmarsh, 2006).

Equation (34) is discretised as

$$\boldsymbol{Cu} = \rho_i g H_0 \frac{\Delta_n \boldsymbol{h}}{\Delta x} - \rho_i g \alpha \boldsymbol{h_u}, \tag{37}$$

where $\boldsymbol{u}$ is the vector of $u_n$; $\boldsymbol{h}$ is the vector of $h_n$; $\boldsymbol{h_u}$ is the vector of thickness $h_n$ averaged to $u$-points; and $\Delta_n(\cdot)$ indicates the first-order finite difference of a discrete function on the numerical grid, i.e. $\Delta_n \boldsymbol{h} = (h_{n+1} - h_n)$. $\boldsymbol{C}$ is a symmetric, tridiagonal matrix with the entries

$$c_{m,n} = \begin{cases} -B - 2D & m = n \\ D & |m-n| = 1 \\ 0 & o.w. \end{cases} \tag{38}$$

where

$$15 \quad D \equiv \frac{4\mu H_0}{\Delta x^2}, \quad B \equiv \frac{3\beta}{3+\eta}. \tag{39}$$

To proceed, an expression for $\boldsymbol{u}$ is needed. For a linear stability analysis of SIA schemes (Hindmarsh, 2001; Cheng et al., 2017), this is straightforward, since velocity depends locally on thickness and surface slope. For the DIVA balance, however, a linear system (Eq. 37) must be solved. Analytical solutions to matrix equations are not always available, but for our assumptions (constant viscosity and flow in one horizontal direction) and geometry (periodic with constant surface slope and thickness), an asymptotically accurate closed-form solution can be derived. The mathematical details can be found in Appendices A1 and A2, but here we state the result:

$$u_m = -\varepsilon_h a_0 e^{ikm\Delta x} \kappa(\Delta x, k)\phi(\Delta x, k, r), \tag{40}$$





where

$$\kappa(\Delta x, k) = \frac{\rho_i g}{4\mu}\Delta x(e^{ik\Delta x} - 1) - \frac{\rho_i g\alpha}{8\mu H_0}\Delta x^2(e^{ik\Delta x} + 1), \tag{41}$$

$$\phi(\Delta x, k, r) = 1 + \frac{re^{ik\Delta x}}{1 - re^{ik\Delta x}} + \frac{re^{-ik\Delta x}}{1 - re^{-ik\Delta x}} = \frac{1 - r^2}{1 + r^2 - 2r\cos(k\Delta x)}, \tag{42}$$

$$r = \left(1 + \frac{q}{2}\right) \pm \sqrt{\left(1 + \frac{q}{2}\right)^2 - 1}, \tag{43}$$

$$a_0 = (2 + q - 2r)^{-1} = \frac{1}{2\sqrt{\left(1 + \frac{q}{2}\right)^2 - 1}}, \tag{44}$$

$$q = B/D = \left(\frac{3\eta}{3 + \eta}\right)\left(\frac{\Delta x}{2H_0}\right)^2. \tag{45}$$

The linearised mass-balance equation, Eq. (35), is discretised with an explicit upwind scheme for the advective term and a centered difference for the divergence term:

$$h_m^{j+1} = h_m^j - u_0\frac{\Delta t}{\Delta x}(h_m^j - h_{m-1}^j) - H_0\frac{\Delta t}{\Delta x}(u_m - u_{m-1}). \tag{46}$$

With $h_m^j = \varepsilon_h e^{ikm\Delta x}$, and using Eq. (40) for $u_m$, this becomes

$$h_m^{j+1} = \varepsilon_h e^{ikm\Delta x}\left[1 - u_0\frac{\Delta t}{\Delta x}(1 - e^{-ik\Delta x}) + H_0\frac{\Delta t}{\Delta x}a_0\kappa(\Delta x, k)\phi(\Delta x, k, r)(1 - e^{-ik\Delta x})\right]. \tag{47}$$

If the real part of the bracketed expression has an absolute value greater than 1 for an initial condition $\varepsilon_h e^{ikx}$, this scheme is unstable.

Writing $\kappa$ in full, the third term in brackets in Eq. (47) (i.e., the divergence term) can be written

$$\Xi = H_0\frac{\Delta t}{\Delta x}a_0\phi(\Delta x, k, r)\left[\frac{\rho_i g}{4\mu}\Delta x(e^{ik\Delta x} - 1) - \frac{\rho_i g\alpha}{8\mu H_0}\Delta x^2(e^{ik\Delta x} + 1)\right](1 - e^{-ik\Delta x})$$

$$= H_0\frac{\Delta t}{\Delta x}a_0\phi(\Delta x, k, r)\left[\frac{\rho_i g}{4\mu}\Delta x(2\cos(k\Delta x) - 2) - \frac{\rho_i g\alpha}{8\mu H_0}\Delta x^2(2i\sin(k\Delta x))\right]. \tag{48}$$

Defining $\theta \equiv k\Delta x$, we seek the value of $\theta$ which maximizes the magnitude of the real part of $\Xi$. In doing so, we effectively ignore the second term in brackets in Eq. (48), which is purely imaginary. This is justified as we are primarily concerned with cases where $\Delta x \to 0$, where this term becomes negligible. In Section 3.4, however, we test the validity of the assumption over a wide range of $\Delta x$ values. Using Eq. (42) to write $\phi(\Delta x, k, r)$ in full, the remaining term is

$$\Xi = -H_0\Delta t a_0\left(\frac{\rho_i g}{2\mu}\right)\left(\frac{1 - r^2}{1 + r^2 - 2r\cos\theta}\right)(1 - \cos\theta). \tag{49}$$

This expression is non-positive, continuously differentiable in $\theta$, and has extremal points at $\theta = 0$ and $\theta = \pi$. The maximum value occurs when $\cos(\theta) = \cos(\pi) = -1$:

$$|\Xi|_{\max} = H_0\Delta t a_0\left(\frac{\rho_i g}{\mu}\right)\left(\frac{1 - r}{1 + r}\right). \tag{50}$$



We refer to the second term in brackets in Eq. (47) as $\gamma$. This term is related to advection and has a maximal value (when $e^{ik\Delta x} = -1$) of

$$|\gamma|_{\max} = 2u_0 \frac{\Delta t}{\Delta x}. \tag{51}$$

Together, $\gamma$ and $\Xi$ determine the stability of the time-stepping scheme. $\gamma$ is related to advection, while $\Xi$ arises from dynamic
thinning and thickening under divergence. The presumption is that an arbitrary initial condition, unless carefully constructed, projects onto the mode $e^{ikx}$ for which Eqs. (50) and/or (51) are realised. The scheme is stable when the real part of the bracketed expression in Eq. (47) has an absolute value less than 1, or equivalently, when $|\gamma| + |\Xi| < 2$. When $\gamma$ and $\Xi$ are of similar magnitude, the associated limit on $\Delta t$ does not have a simple expression. But when one or the other is dominant, there is differing leading-order behaviour in the relationship between resolution and maximal stable timestep.

When $\gamma \gg \Xi$, the equation system is essentially advective, and stability is governed by Eq. (51):

$$\Delta t < \Delta x/u_0 \equiv \Delta t_{adv}. \tag{52}$$

When $\Xi \gg \gamma$, the maximum stable timestep follows from Eq. (50):

$$\Delta t < \left(\frac{2\mu}{\rho_i g H_0}\right) a_0^{-1} \left(\frac{1+r}{1-r}\right) \equiv \Delta t_{\mathrm{dyn}}. \tag{53}$$

The dependence of $\Delta t_{\mathrm{dyn}}$ on $\Delta x$ is more complex than that of $\Delta t_{\mathrm{adv}}$. It is useful, however, to consider two end-member
regimes: large $q$ and small $q$ (in the following, it is assumed unless stated otherwise that $\Delta t_{\mathrm{dyn}} \ll \Delta t_{\mathrm{adv}}$, i.e., advection is not a limiting factor). The case $q \gg 1$ corresponds to high basal friction and/or coarse resolution. In this limit it can be shown (see Appendix A1) that $a_0 \approx r \approx 1/q$, so that $a_0^{-1}(1+r)/(1-r) \approx q$, and Eq. (53) becomes

$$\Delta t < \left(\frac{\mu}{2\rho_i g H_0^3}\right)\left(\frac{3\eta}{3+\eta}\right)\Delta x^2. \tag{54}$$

If $\eta \gg 1$, as in the case of high basal friction, this simplifies to

$$\Delta t < \left(\frac{3\mu}{2\rho_i g H_0^3}\right)\Delta x^2. \tag{55}$$

In this case, the stability criterion is identical to that of the SIA solver (Eq. (29)), with the maximum stable timestep, $\Delta t_{\max}$, proportional to $\Delta x^2$. This condition usually is not very restrictive for the DIVA solver, since large $q$ typically implies coarse resolution if the ice is not too thin.

In contrast, for $q \ll 1$ (i.e., low basal friction and/or fine resolution), it can be shown that $a_0 \approx 1/(2\sqrt{q})$ and $r \approx 1 - \sqrt{q}$.
The terms in Eq. (53) containing $a_0$ and $r$ reduce to $2(2 - \sqrt{q}) \approx 4$, resulting in

$$\Delta t < \frac{8\mu}{\rho_i g H_0}. \tag{56}$$

Thus, in the limit of small $q$ (or equivalently, small $\Delta x$), the time-step limit arising from ice-flux divergence depends on $\mu$ and $H_0$, but not $\Delta x$. This is not to say that the limiting timestep is independent of resolution – but such dependence arises from the





advective term, and thus the maximum stable timestep varies linearly, not quadratically, with $\Delta x$. For a typical ice thickness of $10^3$ m and effective viscosity of $10^7$ Pa y, we would have $\Delta t_{\mathrm{dyn}} \sim 10$ y, which is not very restrictive. The maximum timestep would be determined by the advective limit, Eq. (52). With $\Delta x = 10^3$ m and $u_0 = 10^3$ m yr$^{-1}$, the advective limit would be $\Delta t_{\mathrm{adv}} \sim 1$ yr.

These two regimes of limiting timestep correspond to differing behaviour of the DIVA equations at different scales. In the large-$q$ limit, Eq. (54), variations in velocity correlate closely with driving stress, and the dynamic response behaves like a diffusion process (similar to the SIA) in which the flux of thickness is proportional to thickness gradients, leading to a quadratic dependence of the limiting timestep on resolution. In the small-$q$ limit, Eq. (56), small-scale thickness oscillations are damped by dynamic thinning due to velocity divergence induced by the oscillations. The velocity gradients are scale-independent due

to averaging of associated driving stresses over the membrane scale, resulting in a scale-independent damping rate, and hence a scale-independent timestep limit (provided the advective timestep limit is large enough).

### 3.2   Hybrid solver stability

We consider now the *Hybrid* stress balance, in which sliding velocity is determined by the SSA momentum balance, Eq. (8). Thickness transport is due to a vertically averaged velocity $\bar{u}_{\mathrm{hyb}}$, the sum of sliding velocity and the vertical average of the

SIA velocity from Eq. (6), $\bar{u}_{\mathrm{sia}} = \rho_i g H^2 \alpha / (3\mu)$.

To investigate the Hybrid solver stability, we will first modify the DIVA analysis (Sect. 3.1) to treat the SSA case. Thus, we treat the SSA balance as a first step in the analysis of the Hybrid scheme. The SSA momentum balance (in one dimension, with constant viscosity) is

$$4\mu \frac{d}{dx} \left( H \frac{du}{dx} \right) - \beta u = \rho_i g H \frac{ds}{dx}. \tag{57}$$

This is the low-friction limit of the DIVA balance, Eq. 30, with $u$ replacing $\bar{u}$ since velocity is depth-uniform. The SSA mass balance equation is given by Eq. (32), again replacing $\bar{u}$ with $u$. The RHS of the matrix equation for velocity, Eq. (37), is identical, and the matrix $C$ is modified with $B = \beta$. Thus, the dynamically-limited timestep $\Delta t_{\mathrm{dyn}}$ is given by Eq. (53), but with $r$ and $a_0$ modified by the new definition of $B$. At high resolution, this leads to the same dynamically-limited behaviour as for DIVA, Eq. (56). At coarse resolution, the dynamically-limited timestep differs from DIVA due to the modified frictional

term; we do not give details here.

The Hybrid equations for momentum balance and mass conservation in 1D with constant viscosity $\mu$ are

$$4\mu \frac{d}{dx} \left( H \frac{du_b}{dx} \right) - \beta u_b = \rho_i g H \frac{ds}{dx}, \tag{58}$$

$$\frac{\partial H}{\partial t} = -\frac{\partial}{\partial x} \left( u_b H + \bar{u}_{\mathrm{sia}} H \right) = -\frac{\partial}{\partial x} \left( u_b H - \frac{\rho_i g H^3}{3\mu} \frac{\partial s}{\partial x} \right). \tag{59}$$

We consider the same reference state as in the DIVA analysis. The reference sliding velocity is

$$u_{b,0} = \frac{\rho_i g H_0 \alpha}{\beta}, \tag{60}$$





and the Hybrid depth-averaged velocity in this reference state is

$$\bar{u}_{\mathrm{hyb}} = u_{b,0} + \frac{\rho_i g H_0^2 \alpha}{3\mu} = \rho_i g H_0 \alpha \left( \frac{1}{\beta} + \frac{H_0}{3\mu} \right). \tag{61}$$

Thus, $\bar{u}_{\mathrm{hyb}} = u_0$, the DIVA depth-averaged reference velocity from Eq. (31). As before (cf. Eqs. (34) and (35)), we consider perturbations in $H$ of $\varepsilon_h e^{ikx}$ and write the equations to first order in $\varepsilon_h$:

$$4\mu H_0 \frac{d^2 \delta u_b}{dx^2} - \beta \delta u_b = \rho_i g H_0 \frac{\partial \delta H}{\partial x} - \rho_i g \alpha \delta H, \tag{62}$$

$$\frac{\partial \delta H}{\partial t} = -u_{b,0} \frac{\partial \delta H}{\partial x} - H_0 \frac{d\delta u_b}{dx} + \frac{\rho_i g H_0^3}{3\mu} \frac{\partial^2 \delta H}{\partial x^2} - \frac{\rho_i g H_0^2 \alpha}{\mu} \frac{\partial \delta H}{\partial x}$$
$$= -u_{\mathrm{hyb}}^{\mathrm{eff}} \frac{\partial \delta H}{\partial x} - H_0 \frac{d\delta u_b}{dx} + \frac{\rho_i g H_0^3}{3\mu} \frac{\partial^2 \delta H}{\partial x^2}, \tag{63}$$

where

$$u_{\mathrm{hyb}}^{\mathrm{eff}} \equiv u_{b,0} + \frac{\rho_i g H_0^2}{\mu} \alpha = \rho_i g H_0 \alpha \left( \frac{1}{\beta} + \frac{H_0}{\mu} \right). \tag{64}$$

Since $u_{\mathrm{hyb}}^{\mathrm{eff}} > u_0$ (effectively because there is no '3' in the denominator of the second term in parentheses), the Hybrid advective stability constraint is slightly more restrictive than for DIVA.

Consider the third term on the RHS of Eq. (63), which follows from the dependence of $u_{\mathrm{SIA}}$ on perturbations in $s$. If this term is discretised using an explicit second-order finite difference, it can be expressed as

$$\left( \frac{\rho_i g H_0^3}{3\mu} \right) \frac{h_{m+1} - 2h_m + h_{m-1}}{\Delta x^2} = \left( \frac{\rho_i g H_0^3}{3\mu} \right) \frac{\varepsilon_h e^{ikm\Delta x}}{\Delta x^2} (2\cos(k\Delta x) - 2). \tag{65}$$

Note that this term is missing from the analogous equation for the DIVA solver (Eq. (35)). With this result and those from the previous section (cf. Eq. (47) for DIVA), it can be shown that the limiting timestep is determined by the constraint that

$$\left| 1 - 2u_{\mathrm{hyb}}^{\mathrm{eff}} \frac{\Delta t}{\Delta x} - H_0 \Delta t a_0 \left( \frac{\rho_i g}{\mu} \right) \left( \frac{1-r}{1+r} \right) - \frac{4\rho_i g H_0^3}{3\mu} \frac{\Delta t}{\Delta x^2} \right| < 1. \tag{66}$$

In the above expression, $r$ and $a_0$ are as defined as in Eqs. (43) and (44), but with the substitution $B = \beta$, giving $q = $

$20 \quad \beta \Delta x^2 / (4\mu H_0)$. In the limit $q \gg 1$ (high basal friction and/or coarse resolution), the third term in Eq. (66) reduces to

$$\left( \frac{4\rho_i g H_0^2}{\beta} \right) \frac{\Delta t}{\Delta x^2}.$$

Recalling that $\eta \equiv \beta H_0 / \mu$, the third and fourth terms can be combined to give

$$\left( \frac{4\rho_i g H_0^3}{\mu} \right) \left( \frac{3+\eta}{3\eta} \right) \frac{\Delta t}{\Delta x^2}.$$

Provided the advective timestep limit $\Delta t_{\mathrm{adv}} = \Delta x / u_{\mathrm{hyb}}^{\mathrm{eff}}$ is large, an approximate time step limit in this regime is given by

$$\Delta t < \left( \frac{\mu}{2\rho_i g H_0^3} \right) \left( \frac{3\eta}{3+\eta} \right) \Delta x^2, \tag{67}$$





which is identical to the large-$q$ limit for DIVA, Eq. (54), and reduces to Eq. (55) when $\eta \gg 1$. When $q \ll 1$ (low basal friction and/or fine resolution), the third term in (66) is bounded independently of $\Delta x$. For small $\Delta x$, the timestep limit for small $q$ is therefore governed by the fourth term:

$$\Delta t < \frac{3\mu}{2\rho_i g H_0^3}\Delta x^2.$$

(68)

Since $\Delta t$ is proportional to $\Delta x^2$, Eq. (68) becomes very restrictive at high resolution, like the identical equation for SIA (Eq. (29)).

### 3.3   L1L2-SIA solver stability

As described in Section 2.5, the L1L2-SIA solver consists of a two-step process, the first being a solve for the sliding velocity $u_b$. With a constant viscosity, the resulting equation is identical to that of the Hybrid model, and thus $u_b$ is found by solving

the SSA equation. The depth-averaged velocity is found by depth-integrating the vertical shear stress. In other words, the longitudinal strain rate is approximated by the gradient of $u_b$, and the resulting $x$-momentum balance, given by

$$\partial_x\left(4\mu\frac{\partial u_b}{\partial x}\right) + \partial_z\left(\mu\frac{\partial u}{\partial z}\right) = \rho g\frac{\partial s}{\partial x},$$

(69)

is rearranged and integrated with depth (applying the free-surface boundary condition) to yield

$$\frac{\partial u}{\partial z} = \left(4\frac{\partial^2 u_b}{\partial x^2} - \frac{\rho g}{\mu}\frac{\partial s}{\partial x}\right)(H - z).$$

(70)

This yields the depth-averaged velocity

$$\bar{u}_{\text{L1L2}} = u_b + \left(4\frac{\partial^2 u_b}{\partial x^2} - \frac{\rho g}{\mu}\frac{\partial s}{\partial x}\right)\frac{H^2}{3}.$$

(71)

Since the depth-averaged velocities in the Hybrid and L1L2-SIA approximations differ only by a term involving the second derivative of $u_b$, the reference depth-averaged velocity, $\bar{u}_{\text{L1L2}}$, is again equal to $u_0$. The linearised version of the perturbed mass balance is given by

$$
\begin{aligned}
\frac{\partial \delta H}{\delta t} &= -u_{b,0}\frac{\partial \delta H}{\partial x} - H_0\frac{\partial \delta u_b}{\partial x} - \left(\frac{4H_0^3}{3}\frac{\partial^3 \delta u_b}{\partial x^3} - \frac{\rho_i g H_0^3}{3\mu}\frac{\partial^2 \delta H}{\partial x^2} + \frac{\rho_i g \alpha H_0^2}{\mu}\frac{\partial \delta H}{\partial x}\right) \\
&= -u_{b,0}\frac{\partial \delta H}{\partial x} - H_0\frac{\partial \delta u_b}{\partial x} - \frac{H_0^2}{3\mu}\left(\beta\frac{\partial \delta u_b}{\delta x} + 2\rho_i g\alpha\frac{\partial \delta H}{\partial x}\right) \\
&= -u_{\text{L1L2}}^{\text{eff}}\frac{\partial \delta H}{\partial x} - H_0\left(1 + \frac{\eta}{3}\right)\frac{\partial \delta u_b}{\delta x},
\end{aligned}
$$

(72)

where the replacement of the terms in parentheses in the second equality comes from Eq. (58) and

$$u_{\text{L1L2}}^{\text{eff}} = u_{b,0} + \frac{2\rho_i g H_0^2}{3\mu}\alpha = \rho_i g H_0\alpha\left(\frac{1}{\beta} + \frac{2H_0}{3\mu}\right).$$

(73)

Functionally, Eq. (72) has the same form as (35). The differences are the prefactors on the gradients of perturbed thickness and perturbed velocity, and the fact that the basal (sliding) velocity is considered, and not depth-averaged velocity. Since $u_b$





satisfies the same solution as in the Hybrid balance, the limiting time step is constrained according to

$$\left|1 - 2u_{\text{L1L2}}^{\text{eff}}\frac{\Delta t}{\Delta x} - H_0\left(1 + \frac{\eta}{3}\right)\Delta t a_0\left(\frac{\rho_i g}{\mu}\right)\left(\frac{1-r}{1+r}\right)\right| < 1. \tag{74}$$

Similarly, approximate bounds can be found for the regimes $q \gg 1$ and $q \ll 1$. When $q \gg 1$ the limiting time step can be approximated by

$$\Delta t < \left(\frac{\beta}{2\rho_i g H_0^2(1+\eta/3)}\right)\Delta x^2$$
$$= \left(\frac{\mu}{2\rho_i g H_0^3}\right)\left(\frac{3\eta}{3+\eta}\right)\Delta x^2. \tag{75}$$

This expression is identical to the large-$q$ bounds for DIVA and Hybrid, Eqs. (54), and (67). When $q \ll 1$, the limiting time step asymptotes to

$$\Delta t < \left(\frac{8\mu}{\rho_i g H_0}\right)\left(\frac{3}{3+\eta}\right), \tag{76}$$

which is smaller than Eq. (56), the small-$q$ limit for DIVA, by a factor related to $\eta$. For large $\eta$ (i.e., high basal friction and low sliding), $\Delta t$ is small. But like Eq. (56) for DIVA, Eq. (76) has no resolution dependence. At high resolutions, the maximum stable time step is either independent of $\Delta x$, or, where the advective stability limitation is important, linear in $\Delta x$. Thus the above analysis suggests that, unlike in the Hybrid scheme, the maximum time step does **not** depend quadratically on $\Delta x$. As will be shown, however, simulations carried out with realistic models and/or nonlinear rheology suggest that L1L2-SIA

stability is similar to that of the Hybrid scheme. We explore this contradiction further in the Discussion section.

Table 1 summarizes the low- and high-resolution timestep limits for the DIVA, Hybrid and L1L2-SIA solvers. At low resolution, the limits are identical, since in all three cases the variations in velocity and divergence correlate with those in thickness. At high resolution, the DIVA solver is not dependent on resolution. In the Hybrid balance, however, the SIA component of depth-averaged velocity correlates with thickness variations. As such, the bound on timestep is still quadratic in grid spacing,

but with a larger bounding constant. The L1L2-SIA stability limit is similar to DIVA for high resolution, albeit offset to lower values by the additional term $\frac{3}{3+\eta}$, which is less than 1 and approaches 0 for large $\eta$.

### 3.4 Numerical validation

To confirm the validity of the above analysis, we run numerical simulations with 1D DIVA, Hybrid and L1L2-SIA models derived using the assumptions above. For the 1D DIVA solver, we discretise and solve the simplified equations (30) and (32),

using a first-order upwind finite volume scheme for Eq. (32). We discretise the Hybrid solver analogously using Eqs. (58) and (59), and L1L2-SIA using Eq. (32) with (71) for depth-averaged velocity. These simplified solvers have been implemented in Matlab and are provided along with code to run the tests cases as a Supplement to the paper. As further validation, we run the same simulations using two comprehensive ice sheet models, Yelmo (Robinson et al., 2020) and CISM (Lipscomb et al., 2019). The three solvers have been implemented in Yelmo (Robinson et al., 2020), and all test cases are simulated. CISM has

no Hybrid solver, but is used to test the DIVA and L1L2-SIA solvers.





| Solver | Low resolution | Eq. number | High resolution | Eq. number |
|---|---|---|---|---|
| DIVA | $\Delta t < \frac{\mu}{2\rho_i g H_0^3}\left(\frac{3\eta}{3+\eta}\right)\Delta x^2$ | 54 | $\Delta t < \frac{8\mu}{\rho_i g H_0}$ | 56 |
| Hybrid | $\Delta t < \frac{\mu}{2\rho_i g H_0^3}\left(\frac{3\eta}{3+\eta}\right)\Delta x^2$ | 67 | $\Delta t < \frac{3\mu}{2\rho_i g H_0^3}\Delta x^2$ | 68 |
| L1L2-SIA | $\Delta t < \frac{\mu}{2\rho_i g H_0^3}\left(\frac{3\eta}{3+\eta}\right)\Delta x^2$ | 75 | $\Delta t < \frac{8\mu}{\rho_i g H_0}\left(\frac{3}{3+\eta}\right)$ | 76 |

**Table 1.** Summary of asymptotic timestep limits derived for the DIVA, Hybrid, and L1L2-SIA solvers under simplified conditions (1D, uniform viscosity, infinitely long ice slab of uniform thickness $H_0$, explicit time-stepping scheme). Limits are shown for when the advective limit is sufficiently large so as not to apply, and are defined loosely for "low resolution" and "high resolution" regimes. See the text for details.

The test problem is solved with a periodic uniform slab as described in the beginning of this section, with the constant bedrock slope set to $\alpha = ds/dx = -10^{-3}$. For each model and parameter set, we verified that the diagnosed velocity profile (not shown) is consistent with the exact solutions.

To test stability, we add a random Gaussian perturbation to the initial ice thickness, such that

$$h_n(t=0) = H_0 + \mathcal{X}_n, \quad n = 1, 2, ..N, \tag{77}$$

where $\mathcal{X}_n$ are independent Gaussian random variables with zero mean and standard deviation of 0.1 m. We then run the model forward for 100 timesteps. While not guaranteed, it is likely that $\boldsymbol{h}(t=0)$ will project onto the least stable numerical mode (i.e., that for which the expressions defining $\Delta t_{\mathrm{dyn}}$ and $\Delta t_{\mathrm{adv}}$ are realised).

For a given set of physics parameters $(\mu, H_0, \beta)$, $\Delta x$ is varied over a range from 10 m to 40 km. At each resolution we run

multiple tests with different values of $\Delta t$ to determine the maximum stable timestep. For each run, we calculate the ratio $\sigma/\sigma_0$, where $\sigma$ is the standard deviation of $\boldsymbol{h}$ at the final timestep, and $\sigma_0$ is the standard deviation of $\boldsymbol{h}(t=0)$. The variance in ice thickness should decrease for a stable scheme, so this ratio serves as a metric of stability. We consider $\sigma/\sigma_0 \leq 1$ to indicate stability and $\sigma/\sigma_0 > 1$ to indicate instability. The numerical results are compared to the timestep limits determined analytically above.

We test two cases that are representative of different ice-flow regimes. The first parameter set ($\mu = 1 \times 10^5$ Pa yr, $H_0 = 1000$ m, $\beta = 1000$ Pa m yr$^{-1}$, $\eta = 10$) corresponds to thicker, less viscous ice with a strong bed, i.e., conditions that favor vertical shear over sliding. In this case, $u_0$ is equal to 39 m yr$^{-1}$, and $L_m$ is approximately 1.3 km. With such a low background velocity, the advective timestep limit $\Delta t_{\mathrm{adv}}$ is not restrictive except at extremely high resolution, and so stability is determined by dynamic divergence alone. The second parameter set ($\mu = 4 \times 10^5$ Pa yr, $H_0 = 500$ m, $\beta = 30$ Pa m yr$^{-1}$, $\eta = 0.0375$) cor-

responds to thinner, more viscous ice with a weak bed, i.e., conditions that favor fast sliding and vertically uniform flow. In this case, the maximum stable timestep $\Delta t_{\mathrm{dyn}}$ is generally larger (as a function of $\Delta x$) than in the thick, shearing case. Also, $u_0$ is larger (150 m/a), which means that $\Delta t_{\mathrm{adv}}$ may impose a stability limit as $\Delta x$ becomes small. Figure 1 shows results for the two parameter sets from the simple 1D models and from Yelmo and CISM, as compared to the analytical solutions.





In the case of the DIVA solver, the maximum stable timesteps determined by the 1D model, Yelmo, and CISM are in excellent agreement with the analytical estimate of $\Delta t_{\mathrm{dyn}}$ as given by Eq. (50) for both tests. In the shearing case, the advective timestep limit is large enough that it is not relevant, so the maximum stable timestep transitions from a quadratic dependence on grid resolution for low resolutions ($\Delta x > L_m$) to a constant value. In the sliding case, the maximum stable timestep follows a

similar dependence until $\Delta x < 10^2$ m, at which point the advective limit becomes the limiting factor.

For the Hybrid solver, we again find excellent agreement with the analytical estimate of $\Delta t_{\mathrm{dyn}}$, as given by Eq. (66), for both the simplified 1D model and the ice sheet model Yelmo. In both tests, the maximum stable timestep follows a quadratic dependence on resolution, although a transition occurs to a slightly more stable regime below $\Delta x \sim L_m$ in the sliding case. In the shearing case in particular, the maximum stable timestep drops quickly as $\Delta x$ decreases, such that it is already as small as

$\Delta t \sim 10^{-2}$ when $\Delta x \sim L_m \sim 1000$ m. For the Hybrid solver, the very small timestep limits ensure that the advective timestep limit never becomes relevant in the two tests.

In the case of the L1L2-SIA solver, the simple 1D model agrees with the analytical results, but the comprehensive models do not. According to Eq. (74), at high resolution the L1L2-SIA solver's maximum stable timestep should become independent of grid resolution. The 1D model confirms this result. The maximum stable timestep in the shearing test at high resolution is a

constant value, although offset to a lower value than that of the DIVA solver due to the scaling dependence of the equation on $\eta$. Here as well, for extremely high resolution ($\Delta x < 10$ m), the model becomes unstable in the shearing case – though this may be due to our use of finite perturbations to validate the linear stability analysis, as smaller perturbations allow further filling of the shaded regions at high resolutions (not shown). In contrast to the 1D model, both Yelmo and CISM exhibit a quadratic dependence on grid resolution akin to that of the Hybrid solver for this test. In the sliding test, the 1D model again agrees

with the analytical solution. Yelmo's results again resemble the Hybrid solver here, while CISM maintains a purely quadratic dependence on resolution.

It is not clear why CISM's L1L2-SIA solver is less stable than Yelmo's for the thin sliding case, nor why the theoretical high-resolution stability limit for L1L2-SIA is recovered by the simple 1D model but not by Yelmo and CISM. We point out that the result given by Eq. (76) depends critically on a cancellation of terms in Eq. (72), and it is possible that subtle details of

the finite-difference schemes of these more sophisticated schemes prevent such cancellation. Moreover, it is worth noting that this theoretical result also depends on a constant viscosity, which is not the case in realistic models.

Overall, these experiments show that the stability limits derived under simplified assumptions are valid in numerical simulations. Moreover, they highlight the difference in stability regimes between the DIVA, Hybrid and L1L2-SIA solvers. As shown in the analytical derivation, the SSA solver has the same stability limits as the DIVA solver at high resolution, thus it

can also be expected to be more stable than the Hybrid solver. In contrast, previous work has shown that SIA stability depends quadratically on $\Delta x$ at higher resolutions (Cheng et al., 2017), consistent with the results found for the Hybrid solver that incorporates the SIA solution. While the L1L2-SIA solver appears to have stability limits similar to the DIVA solver in the simplified test setup, the more comprehensive models indicate that the solver may not be so stable in practice.

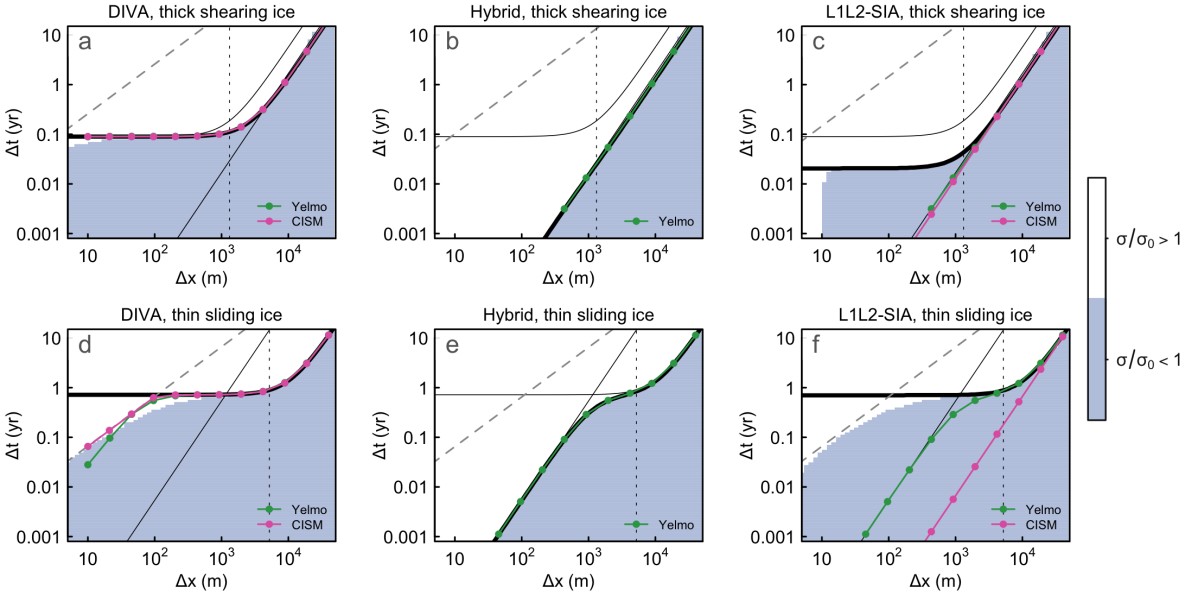

**Figure 1.** Comparison of the analytical stable timestepping regime for the DIVA (a & d), Hybrid (b & e) and L1L2-SIA (c & f) solvers for a 1D slab of ice on a bedrock with constant slope ($\alpha = 10^{-3}$) with thick shearing ice (a–c, $H_0 = 1000$ m, $\mu = 1 \times 10^5$ Pa yr, $\beta = 1000$ Pa m$^{-1}$ yr) and thin sliding ice (d–f, $H_0 = 500$ m, $\mu = 4 \times 10^5$ Pa yr, $\beta = 30$ Pa m$^{-1}$ yr). In all panels, the vertical thin dotted line indicates the membrane-stress length scale $L_m$ and the thin dashed grey line shows the advective timestep limit, $\Delta t_{\mathrm{adv}}$. For reference, the thin solid black lines shows the analytical solutions for an SIA and SSA solver given by Eqs. (29) and (53) with $B = \beta$, respectively (the SIA solution is always depicted as a straight line). The thick black lines show the corresponding analytical solution based on Eq. (53) for DIVA, Eq. (66) for Hybrid, and Eq. (74) for L1L2-SIA. Light blue shading shows where a 1D model following the derivation in the text has a stable solution. The green and magenta lines with solid points show the maximum stable timestep as determined using the ice sheet models Yelmo and CISM, respectively. The Hybrid solver was not implemented in CISM, so results are shown only for Yelmo. In some other panels, the CISM and Yelmo results overlap.

## 4 Greenland experiments

The above analysis sheds light on the mathematical basis for stability differences between approximations in idealised cases. We are most interested, however, in comparing the solvers under more realistic conditions with, for example, a nonlinear rheology. To this end, we perform several quasi-steady-state simulations for the Greenland Ice Sheet (GrIS) at different resolutions. We use the ice sheet model Yelmo (Robinson et al., 2020), which now supports all five solvers (SIA, SSA, Hybrid, L1L2-SIA and DIVA). By using Yelmo for all experiments, we ensure a clean comparison among solvers. We run experiments at four resolutions for each solver: 32 km, 16 km, 8 km and 4 km. The simulations are designed to represent the complexity in typical simulations of real ice sheets. Each simulation starts from the observed GrIS geometry and uses present-day boundary forcing as in, e.g., the initMIP experiments (Goelzer et al., 2018; Seroussi et al., 2019), with a non-linear rheology following Glen's





flow law with $n = 3$. The basal hydrology and thermodynamics are interactive, with a linear basal friction law that depends on the basal water layer thickness. The exact configuration is not critical to the analysis, as similar behavior is observed under a variety of conditions and domains.

To save computation time, the simulations are not run fully to steady state. Rather, we run the model for a total of 3 kyr. For
the first 2 kyr, we briefly spin up the thermodynamics and basal hydrology and let the model reach a self-consistent state with the topography. We then evaluate the timestep stability during an additional 1 kyr simulation.

Yelmo uses an adaptive timestepping scheme that determines the optimal timestep for the current model conditions (Robinson et al., 2020; Cheng et al., 2017). Given a measure of the truncation error $\tau_t$ in ice thickness at each timestep, the scheme ensures that the maximum value of the truncation error over the domain $\eta_t = \max(\tau_t)$ remains, on average, within a given
tolerance level, $\varepsilon_t$. The timestep is reduced when $\eta_t > \varepsilon_t$ and increased when $\eta_t < \varepsilon_t$, leading to a value that is as large as possible while maintaining stability. In the following analysis, the mean adaptive timestep over the final 1 kyr of the simulations is used as a metric of the model's computational performance. This metric can be compared to the analytically derived values in the simpler 1D formulation in Section 3.

First, it is instructive to compare the simulated GrIS using each solver, given the same boundary conditions and other model
settings (Fig. 2). For the three solvers that include both shear and membrane stresses (Hybrid, L1L2-SIA, and DIVA), the surface velocity fields are similar. All three schemes capture the balance between the gravitational driving stress and vertical shear stress in the interior, between driving stress and membrane stress in ice shelves, and among all three kinds of stresses in fast-sliding ice streams. The ice thickness and velocity distributions are nearly indistinguishable at the regional and large scale. In contrast, the SIA solver simulates slow inland velocities well, but margin velocities are markedly reduced compared to the
solvers with membrane stresses. Since the overall velocity field is dominated by rather slow flow ($< 1000\,\mathrm{m\,yr^{-1}}$), the SIA solver generates a reasonable solution for Greenland, but would clearly fail for, e.g., floating ice. Meanwhile, the SSA solver does not account for vertical shear and thus fails to reproduce inland velocities. The SSA solver can be tuned to reproduce present-day velocities well with the proper optimization of basal friction (e.g., Goelzer et al., 2018), but it is not designed for this regime a priori. For purposes of this analysis, the Hybrid, L1L2-SIA, and DIVA solvers produce the most realistic velocity
fields, but it is useful to include the SIA and SSA solutions as representative of the two extreme flow regimes (pure shearing and pure sliding).

An analysis of model stability, illustrated in Fig. 3, shows large, systematic differences among solvers. Two solver families emerge: the "SIA" and "SSA" families. For all solvers, the maximum stable timestep decreases nonlinearly with increasing grid resolution (Fig. 3a), as shown by the fitted exponent $p$ in the relation $\Delta t \sim \Delta x^p$. Such a relationship could be expected
from the stability analysis in Section 3, since the simulations here correspond more closely to the "low-resolution" regime highlighted in Table 1. The three solvers that rely on the SIA in some form (SIA, Hybrid and L1L2-SIA) have similarly high values of $p$, $\approx$ 2.5–2.8, and thus a large decrease in timestep with finer resolution. In contrast, the SSA and DIVA solvers have a much weaker dependence on grid resolution ($p \approx$ 1.5–1.6), resulting in larger stable timesteps. At the coarsest resolution, $\Delta x = 32\,\mathrm{km}$, all solvers allow a timestep of a similar order of magnitude, between 1–5 yrs, although the SIA family requires
timesteps more than two times smaller than the SSA family. As the resolution increases to $\Delta x = 4\,\mathrm{km}$, the SIA family requires



timesteps of less than $\Delta t = 0.01$ yr, while the SSA and DIVA solvers remain stable for timesteps at least an order of magnitude larger.

We also tested solver performance for a constant timestep, to ensure that the adaptive timestepping scheme provides good estimates of the maximum stable timestep. We ran simulations using only the Hybrid, L1L2-SIA and DIVA solvers for the GrIS domain at 8 km resolution with fixed timesteps of $\Delta t = 0.05$, $0.1$ and $0.5$ yr. From the simulations with adaptive timestepping, we expect that the three solvers would be stable at $\Delta t = 0.05$ yr, but that the Hybrid and L1L2-SIA solvers would become unstable for larger timesteps. Figure 4 confirms that this is the case. For the lowest timestep of $\Delta t = 0.05$ yr, all solvers complete the simulation successfully, with a similar model speed of $\sim 0.2$ kyr per computation hour. At timesteps of $\Delta t = [0.1, 0.5]$ yr, the Hybrid and L1L2-SIA solvers crash early in the simulation, while the DIVA solver completes the simulations successfully. A DIVA simulation with $\Delta t = 0.5$ yr runs approximately 10 times faster than a simulation with $\Delta t = 0.05$ yr.

The difference in maximum stable timestep between solvers results in a marked difference in computational speed (Fig. 3b). At lower resolutions, the differences are modest. The SSA and DIVA solvers run about twice as fast as the Hybrid and L1L2-SIA solvers, while the SIA solver is on par with the fastest solvers because it is computationally much less intensive. As the resolution increases, the timestep dependency on resolution becomes the limiting factor, and the solvers again separate into the faster SSA family and the slower SIA family.

With the exception of the SIA solver, all solvers are comparable in terms of computational time per timestep. Ordered from fastest to slowest per timestep, the solver ranking is SIA, SSA, DIVA, Hybrid, L1L2-SIA. The SIA solver does not require any matrix solutions, so it is much less expensive. The SSA solver requires a computationally intensive matrix solution, but does not require additional calculations for the 3D horizontal velocity field due to the assumption of no vertical shear. The Hybrid, L1L2-SIA and DIVA solvers use the same 2D matrix-solution method as the SSA solver, and all require some vertical integration as well. The L1L2-SIA solver requires more intermediate calculations than the Hybrid or DIVA solvers, but for a Glen's flow law exponent of $n = 3$, as is the case here, an additional iterative step to determine the effective viscosity for the L1L2-SIA solver can be avoided via an analytical solution. Further analysis (not shown) indicates that all solvers require a similar number of Picard iterations to arrive at the converged matrix solution. This shows that the model timestep is the primary determinant of overall model speeds.

Taking DIVA as a reference, we can benchmark its performance against the other solvers (Fig. 3c). The DIVA solver is comparable in computational performance to the SSA solver at all resolutions, although SSA is systematically faster due to a somewhat larger stable timestep and fewer operations per timestep. The performance advantage of DIVA over the other solvers increases with resolution. DIVA runs about two times faster than the Hybrid and L1L2-SIA solvers with $\Delta x = 32$ km, and about 15 times faster at $\Delta x = 4$ km. Interestingly, DIVA also runs faster than the simpler SIA solver at high resolution, reaching speeds up to five times faster at $\Delta x = 4$ km. It may be that the numerical implementation of Yelmo's SIA solver can be improved, but in its current form, the lower computational demand per timestep is not sufficient to compensate for a timestep nearly two orders of magnitude smaller than the DIVA timestep. Furthermore, the SIA solver does not resolve the same complexity of ice-flow physics. This is also true of the SSA solver, which does perform marginally better than the DIVA solver, but cannot represent all large-scale ice-flow regimes.



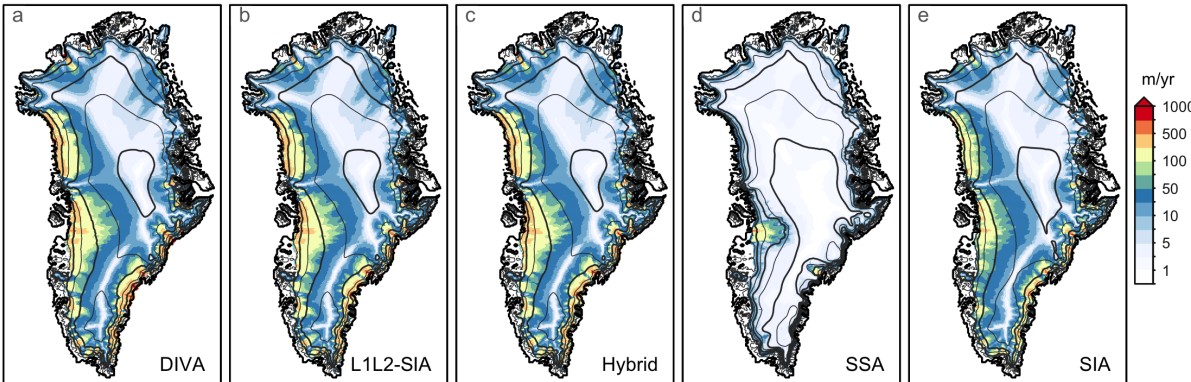

**Figure 2.** Simulated surface velocity (m yr$^{-1}$) at 8 km resolution at the end of the 3-kyr simulation using the (a) DIVA, (b) L1L2-SIA, (c) Hybrid, (d) SSA and (e) SIA solvers as implemented in the Yelmo ice sheet model.

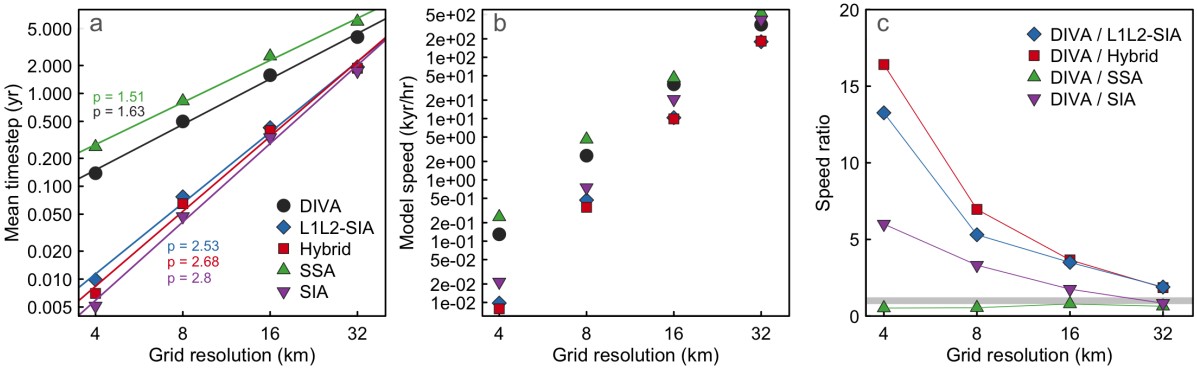

**Figure 3.** (a) Mean model timestep, (b) mean model speed (kiloyears model time per hour of computation on one processor) and (c) ratio of the DIVA mean model speed relative to other solvers (a ratio of $> 1$ implies the simulation using the DIVA solver ran faster) versus grid resolution.

## 5   Discussion

We have shown that different stress-balance approximations can be subject to different stability constraints that are not imme-
diately apparent. The analytical stability analysis in Sect. 3 showed that at low resolution (i.e., $\Delta x > 5 - 10$ km), all solvers are
subject to a diffusive-like quadratic dependence on grid resolution. Strictly speaking, the stability limit depends on $q$, which
is a function of $\eta = \beta H_0 / \mu$ and $H_0$ as well as $\Delta x$. However, resolution is the dominant factor in the tests performed here.
At high resolution, two solver families emerge: those whose dynamic timestep limit becomes independent of grid resolution
(as shown for the SSA and DIVA solvers) and those whose timestep limit maintains the quadratic dependence. These results
were derived for simplified conditions (1D flow with uniform viscosity and constant slope), but the analysis is consistent with
Greenland simulations performed without these simplifications.

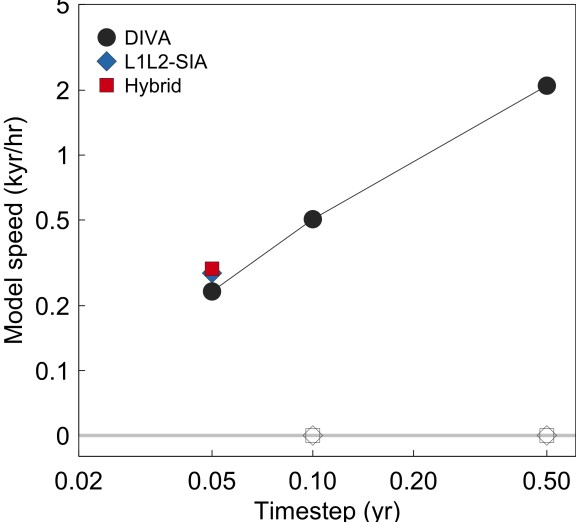

**Figure 4.** Computational speed (kiloyears model time per hour of computation on one processor) versus prescribed timestep for simulations of the GrIS at 8 km resolution using the DIVA (black circlces), L1L2-SIA (blue squares) and Hybrid (red triangles) solvers. Empty symbols at a speed of zero indicate simulations that have crashed.

The L1L2-SIA solver is a notable exception, in that a simplified analysis with constant viscosity indicated stability akin to the DIVA solver, albeit offset to a lower maximum timestep limit at high resolution. Moreover, a simplified numerical model based on these assumptions was consistent with the analysis. However, when using comprehensive ice sheet models for either the simplified experiments or the more realistic Greenland experiments, the stability of the L1L2-SIA solver was more consistent
with the Hybrid solver. Cornford et al. (2013) noted that using the reconstructed depth-averaged L1L2-SIA velocity in their mass continuity scheme leads to instability as well, and they resort to using only the basal sliding velocity to evolve thickness. Thus it appears to be challenging to implement the L1L2-SIA solver (which is indeed more complex to implement than the other solvers) in a way that retains the more stable timestepping behavior.

The results of the GrIS experiments confirm the emergence of the two solver families, with the SIA and SSA solvers serving
as extreme bounds. The SSA and DIVA solvers show a reduced (less than quadratic) dependence on grid resolution as the grid is refined. The SSA solver maintains a systematically larger timestep than DIVA, most likely because vertical shear stress does not contribute to a reduction in stability. In contrast, the SIA, Hybrid and L1L2-SIA solvers show a strong resolution dependence, with p-values in the relation $\Delta t \sim \Delta x^p$ ranging from 2.5–2.8. The SIA solver requires a systematically lower timestep than the Hybrid and L1L2-SIA solvers, likely because the driving stress is not dissipated in any way via membrane
stresses as in the other two solvers. Analogous simulations for the Antarctic Ice Sheet (not shown), which has large, fast-flowing ice shelves, give similar results as shown here for the GrIS.

Continental-scale simulations using Yelmo at higher resolutions than $\Delta x = 4$ km were not attempted, but the analytical results indicate that a further performance advantage of DIVA could be expected. At such high resolution, factors other than





the dynamics, such as basal hydrology and thermodynamics, may play limiting roles in the maximum stable timestep. Also, at a high enough resolution, the advective limit will further restrict the timestep for the DIVA solver.

All solvers were tested in Yelmo with the same numerical treatment of mass conservation. This served to compare the solvers on an equal basis. However, some of the timestep limitations presented here might be alleviated by the use of mass conservation
schemes tailored to the solver in question. For example, applying an implicit scheme to the SIA contribution to velocity can improve stability (Bueler and Brown, 2009). Nonetheless, specific schemes tailored to the choice of dynamics solver, including fully implicit approaches, come with their own tradeoffs and limit the flexibility of the model.

Based on our analysis, the key difference in performance between the two families of solvers emerges in ice-flow regimes that are predominantly driven by vertical shear stress. This is consistent with results of the ISMIP-HOM experiments (Pattyn
et al., 2008), which compare the ability of different solvers to reproduce expected features of ice dynamics in an idealized setting. Previous work has shown that, when membrane stresses dominate (experiment C of Pattyn et al., 2008), the DIVA, L1L2-SIA, and SSA solvers compare well to the benchmark Stokes solutions for a range of length scales (Pattyn et al., 2008; Goldberg, 2011; Perego et al., 2012). However, when sliding is not permitted and shear stress plays a stronger role (experiment A), the results from all three solvers deviate from those of Stokes solvers as the length scale decreases (Fig. 5). The Hybrid and
L1L2-SIA solvers perform quite poorly, as expected, since without sliding, velocities are only represented by the SIA solution. The DIVA solver gives results closer to the Stokes solutions, but with decreasing fidelity at smaller length scales (Goldberg, 2011; Lipscomb et al., 2019). From this, we can understand that the solvers that are less stable numerically as resolution increases (those that reduce to SIA) are also those whose representation of ice-flow physics is least robust.

In general, it should be noted that the simple slab test presented here (Sec. 3) serves as an excellent benchmark for testing
the maximum stable timestep of an ice-sheet model formulation. It is computationally cheap, avoids complications related to lateral boundaries (e.g., calving) and is simple to implement. Most importantly, our results show that the relationship of the maximum stable timestep versus resolution determined via this test should be representative of model stability in more realistic experiments.

## 6   Conclusions

We have investigated the numerical performance of several commonly used ice-dynamics solvers. We focused on three fast, depth-integrated solvers that permit continental-scale simulations of ice-sheets, namely the Hybrid, L1L2-SIA, and DIVA solvers. These solvers treat both shear and membrane stresses, so they are appropriate for simulating large-scale ice sheets. We included the SIA and SSA solvers as useful boundary cases that treat only shear or membrane stresses, respectively.

As a first step, we derived expressions for maximum stable timesteps for the DIVA, Hybrid, and L1L2-SIA solvers in an
idealized 1D configuration. This analysis showed that with coarse resolution, the maximum stable timestep in all three solvers is proportional to the grid resolution squared (i.e., $\Delta t \sim \Delta x^2$). For high resolution, the timestep limit in the Hybrid solver maintains this resolution dependence, greatly restricting the timestep. In contrast, this term drops out for the DIVA solver, and the maximum stable timestep (assuming that stability is not limited by the advective timestep) depends on ice thickness and





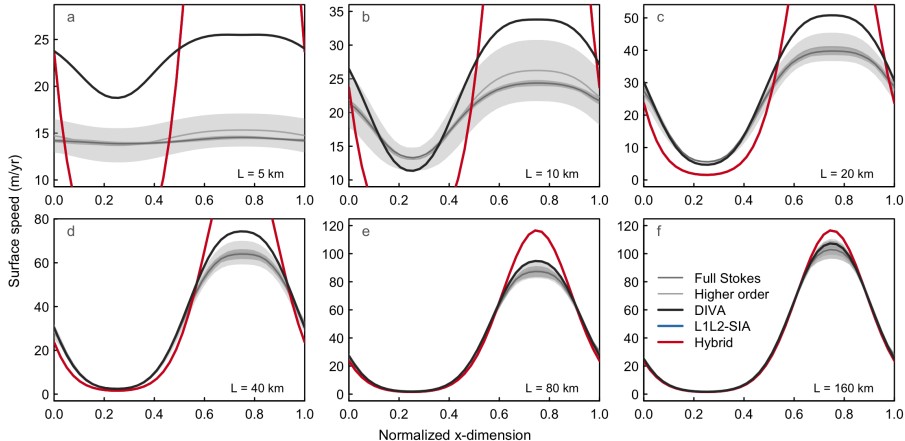

**Figure 5.** Comparison of velocity solutions for the DIVA (black lines), L1L2-SIA (blue lines) and Hybrid (red lines) solvers as implemented in Yelmo to the Full Stokes (dark grey lines and bands) and Higher-order (light grey lines and bands) solvers used in the ISMIP-HOM benchmark experiment A (ice flow over a bumpy bed with zero sliding; Pattyn et al. 2008) for six length scales (5, 10, 20, 40, 80 and 160 km) in panels a-f, respectively. The L1L2-SIA and Hybrid solutions are identical, both reducing to the local SIA solution in the absence of sliding (thus the lines overlap).

viscosity. The result is that the DIVA solver may use a larger timestep than the Hybrid solver, especially at higher resolutions. The stability analysis shows that the SSA solver alone has similar stability characteristics to the DIVA solver. The analysis suggests that L1L2-SIA stability should be similar to DIVA, but tests using Yelmo and CISM showed that its numerical stability is closer to that of the Hybrid solver.

We next performed quasi-steady-state simulations of the GrIS using the five solvers for grid resolutions ranging from 4–32 km. We found that two families of solvers emerge, largely consistent with the analytical results. The SSA and DIVA solvers are stable for larger timesteps, with a less than quadratic dependence on grid resolution. In contrast, the SIA, Hybrid, and L1L2-SIA solvers show reduced stability at high resolution, with an overall more than quadratic dependence on $\Delta x$. Again, the L1L2-SIA solver falls into the latter family.

Overall, our analysis shows the DIVA solver to be superior to the Hybrid and L1L2-SIA solvers for reasons of greater numerical stability as resolution increases, and preferable to the SIA and SSA solvers because of greater physical fidelity in different parts of the ice sheet. Its representation of the stress balance is consistent with full Stokes solutions over a range of length scales. As continental-scale simulations are performed at higher resolutions, the Hybrid and L1L2-SIA solvers may become bottlenecks for model performance, while the DIVA solver remains computationally efficient.

*Code availability.* Yelmo is maintained as a git repository hosted at https://github.com/palma-ice/yelmo under the licence GPL-3.0. Model documentation can be found at https://palma-ice.github.io/yelmo-docs/. The exact version of the model, along with the necessary input data,





used to produce the results used in this paper is archived on Zenodo (https://www.xyz) and has been tagged in the repository as v1.XX. CISM

is an open-source code developed on the Earth System Community Model Portal (ESCOMP) Git repository at https://github.com/ESCOMP/CISM.

https://www.overleaf.com/project/60b11104d5d6cd5fefcd9940

**A1: Velocity derivation in stability analysis**

In this appendix we derive Eq. (40) for $\boldsymbol{u}$, the solution to Eq. (37). We rescale this linear system as

$$\mathbf{A}\boldsymbol{u} = -\frac{\rho_i g}{4\mu}\Delta x \Delta_n \boldsymbol{h} + \frac{\rho_i g \alpha}{4\mu H_0}\Delta x^2 \boldsymbol{h_u} \tag{78}$$

where the elements of $\mathbf{A}$ are

$a_{m,n} = \begin{cases} 2+q & m=n \\ -1 & |m-n|=1 \\ 0 & o.w. \end{cases}$

where $q$ is given by

$$q = B/D = \frac{3\eta}{3+\eta}\left(\frac{\Delta x}{2H_0}\right)^2. \tag{79}$$

Thus, $q$ is non-negative and is determined by two non-dimensional parameters, $\eta = \beta H/\mu$ and $\Delta x/H_0$. Here, it is understood

that $\mathbf{A}$ is *circulant*. That is, each row is displaced by one element to the right compared to the row above, with $a_{N,1} = a_{1,N} =$

$-1$,

  The inverse of $\mathbf{A}$, denoted by $\mathbf{A}^{-1}$, has an approximate analytic expression $\mathbf{G}$ (derived in appendix A2) where the approxi-

mation improves exponentially the larger the size of the domain $L$ relative to $L_m$, as defined in Eq. (36). The element on row

$m$ and column $n$ of $\mathbf{G}$ can be written as

$$g_{m,n} = a_0 r^{\mathscr{L}(m,n,N)} \tag{80}$$

where

$$r = 1 + \frac{q}{2} - \sqrt{\left(1+\frac{q}{2}\right)^2 - 1}, \tag{81}$$

$$a_0 = (2+q-2r)^{-1} = \frac{1}{2\sqrt{\left(1+\frac{q}{2}\right)^2-1}}, \tag{82}$$





and

$$\mathscr{L}(m,n.N) = \min(|m-n|, |m+N-n|, |m-N+n|). \tag{83}$$

That is, $\mathscr{L}$ is the closest "distance" between $m$ and $n$ accounting for periodicity. Note that $q$ uniquely determines the elements of $\mathbf{G}$. For large $q$, it can be shown that $a_0 \approx r \approx 1/q$, and for small $q$, we have $a_0 \approx 1/(2\sqrt{q})$ and $r \approx 1 - \sqrt{q}$.

For any $q > 0$, we have $0 < r < 1$. Thus, for matrix elements far from the main diagonal (i.e., with sufficiently large $M \equiv |m-n|$), $g_{m,n}$ becomes negligible compared to $a_0$. Suppose we define "negligible" as smaller than $a_0 e^{-p}$ for some $p > 0$. Then the condition for element $g_{m,n}$ to be negligible is $r^M < e^{-p}$, or equivalently, $M \ln r < -p$. For small $q$, where $r \approx 1 - \sqrt{q}$, the condition for negligible matrix elements (recalling that $\ln(1-x) \approx -x$ for small $x$) becomes

$$M\sqrt{q} \gtrsim p. \tag{84}$$

Comparing Eq. (36) and Eq. (79), we find $\sqrt{q} = \Delta x / L_m$, and therefore Eq. (84) can be written as $M\Delta x \gtrsim pL_m$. Thus, $L_m$ can be interpreted as a length scale over which the elements of the matrix inverse become small, and the condition $L_m \ll L$ ensures that entries far from the diagonal approach zero. Moreover, the expression for $\mathbf{G}$ shows that velocity, as obtained by inverting Eq. (78), is a *localised weighted average* of driving stress (i.e., the right hand side of Eq. (78)), with $L_m$ as a measure of the "averaging kernel".

We refer to the right hand side of Eq. (78) as $\boldsymbol{R}$, with elements $R_n$. Recall that thickness and velocity points are staggered, with $h_{un}$ located between $h_n$ and $h_{n+1}$, so that $\Delta_n h = h_{n+1} - h_n$, and $h_{un} = (h_{n+1} + h_n)/2$. With $\delta H$ as in (33), $R_n$ is given by

$$R_n = -\varepsilon_h e^{ikn\Delta x}\left[\frac{\rho_i g}{4\mu}\Delta x(e^{ik\Delta x} - 1) - \frac{\rho_i g\alpha}{8\mu H_0}\Delta x^2(e^{ik\Delta x} + 1)\right]$$
$$\equiv -\varepsilon_h e^{ikn\Delta x}\kappa(k,\Delta x), \tag{85}$$

where for compactness we have written the large bracketed expression as $\kappa(k,\Delta x)$. Pre-multiplying Eq. (78) by $\mathbf{G}$, the velocity
$u_m$ can be computed as

$$u_m = \sum_{n=1}^{N} g_{m,n} R_n$$
$$= -\varepsilon_h a_0 \kappa(k,\Delta x)\sum_{n=1}^{N} r^{\mathscr{L}(m,n.N)} e^{ikn\Delta x}, \tag{86}$$

where the sum is taken over all columns of $\mathbf{G}$ with non-negligible entries in row $m$. The series in Eq. (86) can be written as a sum of a constant term (i.e., 1) and two series – one corresponding to $n > m$ and the other to $n < m$. Without loss of generality, we assume $0 \ll m \ll N$ and replace $\mathscr{L}(m,n.N)$ by $|m-n|$. Further, we assume that $N$ is large enough that the terms in the
series are negligible for large $|m-n|$. Thus, the two series ($n > m$ and $n < m$) can be viewed as infinite geometric sums of the form $z(1 + z + z^2 + ...)$, where $z = re^{ik\Delta x}$ for the first series and $z = re^{-ik\Delta x}$ for the second series. These infinite series



will converge to $z/(1-z)$ provided that $|z| < 1$, which follows from $0 < r < 1$ as shown above. This results in the expression given in Sect. 3.1, Eq. (40):

$$u_m = -\varepsilon_h a_0 \kappa(k, \Delta x) e^{ikm\Delta x} \left[ 1 + \frac{re^{ik\Delta x}}{1 - re^{ik\Delta x}} + \frac{re^{-ik\Delta x}}{1 - re^{-ik\Delta x}} \right]$$
$$\equiv -\varepsilon_h a_0 \kappa(k, \Delta x) e^{ikm\Delta x} \phi(\Delta x, k, r), \tag{87}$$

where we have defined a function $\phi$ that can be simplified and shown to be real:

$$\phi(\Delta x, k, r) = 1 + \frac{re^{ik\Delta x}}{1 - re^{ik\Delta x}} + \frac{re^{-ik\Delta x}}{1 - re^{-ik\Delta x}}$$
$$= \frac{1 - r^2}{1 + r^2 - 2r\cos(k\Delta x)}. \tag{88}$$

**A2: Approximate matrix inverse $G$**

The matrix $\mathbf{A}$ in Eq. (78) is a circulant matrix with diagonal terms $2 + q$ and first off-diagonal terms $-1$, and zero elsewhere. Define by $a_{m,n}$ the entry at row $m$ and column $n$ of $\mathbf{A}$, and $a_{m,n}^{inv}$ the entry at row $m$ and column $n$ of $\mathbf{A}^{-1}$. Since $\mathbf{AA}^{-1} = \mathbf{I}$, the identity matrix, we require that for each row $m$, the following must hold:

$$-a_{m,m-1}^{inv} - a_{m,m+1}^{inv} + (2+q)\, a_{m,m}^{inv} = 1, \tag{89}$$

and also that the inner product of row $m$ of $\mathbf{A}^{-1}$ with any column $n$ of $\mathbf{A}$, $m \neq n$, is zero, i.e.

$$-a_{m,n-1}^{inv} - a_{m,n+1}^{inv} + (2+q)\, a_{m,n}^{inv} = 0. \tag{90}$$

We do not derive an analytical expression for a matrix that satisfies Eqs. (89) and (89) for all $m$ and $n$ (and we do not know that one exists). Rather, we derive here an analytically defined matrix $\mathbf{G}$ which is *close* to $\mathbf{A}^{-1}$ in the sense that $(\mathbf{GA} - \mathbf{I})$ is small. We choose an ansatz $\mathbf{G}$ as follows:

$$g_{m,n} = a_0 r^{\mathscr{L}(m,n,N)} \tag{91}$$

where $\mathscr{L}(m.n.N) = \min(|m-n|, |m+N-n|, |m-N+n|)$. That is, $\mathbf{G}$ is a circulant matrix and $\mathscr{L}$ is the "distance" in columns between $m$ and $n$, but accounting for the circulant property of $\mathbf{G}$. For $m \neq n$, Eq. (90) yields

$$a_0 r^{-1} r^{\mathscr{L}(m,n,N)}\left( -r^2 + (2+q)r - 1 \right) = 0, \tag{92}$$

yielding the solution

$$r = \left(1 + \frac{q}{2}\right) \pm \sqrt{\left(1 + \frac{q}{2}\right)^2 - 1}. \tag{93}$$

We take only the negative branch of the solution, leading to Eq. (81). Although this choice is not immediately apparent, it should become clear below. In particular, Eq. (90) is not satisfied everywhere by $\mathbf{G}$. When $\mathscr{L}(m,n,N) = \lfloor N/2 \rfloor$ (where $\lfloor \cdot \rfloor$




indicates the floor function), the inner product of $\mathbf{G}_m$, the $m$th row of $\mathbf{G}$, and $\mathbf{A}_n$, the $n$th column of $\mathbf{A}$, is

$$a_0\left(-r^{N/2-1}+(2+q)r^{N/2}-r^{N/2-1}\right)=a_0 r^{N/2}\left(2+q-2/r\right) \tag{94}$$

if $N$ is even, and

$$a_0\left(-r^{\lfloor N/2\rfloor-1}+(2+q)r^{\lfloor N/2\rfloor}-r^{\lfloor N/2\rfloor}\right)=a_0 r^{\lfloor N/2\rfloor}\left(1+q-1/r\right) \tag{95}$$

if $N$ is odd. We are interested in the limit of high resolution (i.e., small $q$) and its influence on stability. As discussed in A1, as $q$ goes to zero, we have $r\approx 1-\sqrt{q}$, and therefore (using $\log(1-x)\approx -x$ for small $x$):

$$-\log r\sim\sqrt{q}=\frac{\Delta x}{L_m},$$

from which it can be shown that $r^{N/2}$ is asymptotic to $e^{-p}$, where $p=L/(2L_m)$. (Note that if the positive branch of Eq. (93) were taken, the non-zero off-diagonal entries of $\mathbf{GA}$ would instead grow without bound.) Thus, as long as the numerical
domain is sufficiently large compared to $L_m$, the off-diagonal terms of the matrix product $\mathbf{GA}$ (and equivalently $\mathbf{AG}$) are bounded by $e^{-p}$, where the correct choice of $a_0$ (given below) ensures that the diagonal entries are 1. It remains to find $a_0$. Eq. (89) becomes

$$a_0\left(2+q-2r\right)=1, \tag{96}$$

yielding Eq. (82). It can be seen numerically (Fig. 6) that the rows of $\mathbf{G}$ are a good approximation to those of $\mathbf{A}^{-1}$.

*Author contributions.* A.R. conceived of the study. D.G. derived and performed the analytical stability analysis with input from W.H.L. A.R. performed the experiments with Yelmo, and W.H.L. ran the experiments with CISM. All authors contributed to the analysis of the results and writing of the manuscript.

*Competing interests.* The authors declare that they have no competing interests.

*Acknowledgements.* Alexander Robinson was supported by the Ramón y Cajal Programme of the Spanish Ministry for Science, Innova-
tion and Universities (grant no. RYC-2016-20587), as well as the Spanish Ministry of Science and Innovation project ICEAGE (grant no. PID2019-110714RA-I00). Daniel Goldberg was supported by the Natural Environment Research Council (grants NE/S006796/1 and NE/T001607/1). This material is based upon work supported by the National Center for Atmospheric Research, which is a major facility sponsored by the National Science Foundation under Cooperative Agreement No. 1852977.



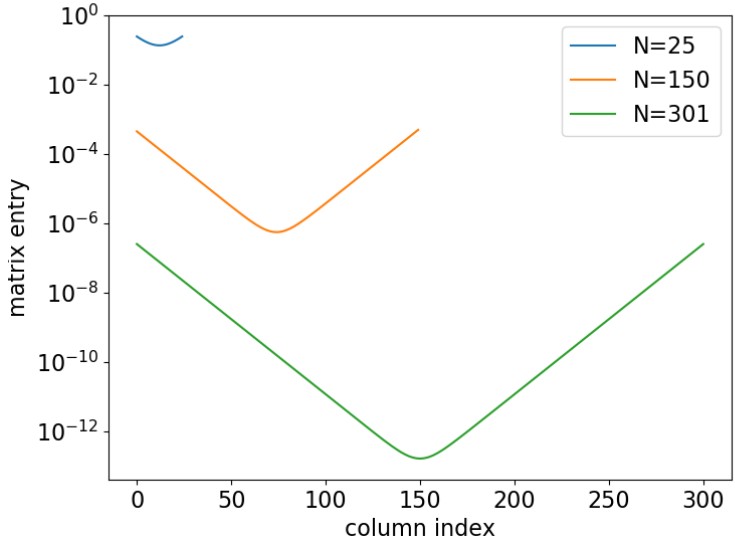

**Figure 6.** Difference between coefficients of row $\lfloor N/2 \rfloor$ of matrix $\mathbf{G}$ and that of $\mathbf{A}^{-1}$ for $q = 0.001$.

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
