# Peer review of "A comparison of the performance of depth-integrated ice-dynamics solvers"

_The Cryosphere, 2021_

## Author Comment (AC1)

To the Editor and reviewers,

Thank you for the detailed and helpful comments that have improved the manuscript significantly.

We would like to point out several major changes made during the revision. First, a Hybrid solver has now been implemented in CISM, and several bug fixes were made to CISM's L1L2-SIA solver. The former agrees well with the Yelmo and 1D results, while the latter also shows closer agreement with Yelmo.

We also revised the stability analysis of the L1L2-SIA solver. Reviewer 1 expressed concern over the disagreement between the theoretical stability analysis and the results of Yelmo and CISM, and also cautioned against the "**reformulations**" at the level of the continuous equations, which were done in Sect. 3.3 of the previous draft to enable such a stability analysis. Analysing section 3.3 more carefully, we realised that a term relating to the surface no-stress boundary condition had been omitted in the formulation of the depth-averaged L1L2-SIA velocity in the 1D model. Further analysis showed that to make headway, even more reformulations would be necessary.

We decided that these reformulations were too heuristic in nature, and that any results they produced, *even if borne out by the 1D model*, should not be presented as rigorous results. **Thus, we no longer present an analytical stability bound for L1L2-SIA.** Instead, we now state at the outset that a rigorous expression for an L1L2-SIA stability bound is beyond the scope of this study, simplifying Sect. 3.3. Alternatively, we propose a heuristic argument, which is examined through experiments in Sect. 3.4, to explain the differing stability behaviour across the L1L2-SIA solvers. In Table 1 now, no entry is given for L1L2-SIA, and in Figs. 1c and 1f, no curves are shown for L1L2-SIA analytical stability bounds.

We believe that these major changes improve the clarity of the manuscript while also addressing the reviewers' general concerns. Below, we have responded to each point individually, with corresponding modifications to the text.

**Reviewer 1 (Mauro Perego)**

In this paper the authors investigate the stability and performance of well known depth-integrated ice sheet models (SSA, SSA+SIA, L1L2, DIVA). The stability is studied theoretically on 1d problems with the assumption of uniform viscosity, and for explicit time integration schemes. The theoretical results are confirmed, for the most part, by numerical experiments. The authors also compare the stability and performance of these different models using the ice sheet codes CISM and Yelmo for modeling the Greenland ice sheet. Among these models, DIVA has better stability properties, and consequently performs better.

Overall the paper is well written and I think its contribution is very important. The theoretical and numerical stability results provided therein will certainly guide modelers and developers in choosing what depth-integrated models to use/implement.

I'm concerned about the discrepancy of their stability analysis and the results obtained using the L1L2 model implemented in CISM and Yelmo. I think it might have to do with the peculiar discretization chosen -see detailed review below- and I wonder whether they could get a better agreement if they changed the discretization used in the analysis.

**We appreciate the reviewer's concerns. As mentioned above, we decided not to present a formal stability analysis for L1L2. We discuss this concern in more detail below.**

I also suggest that the authors develop the theoretical analysis in an unbounded domain, which would make it simpler and cleaner.

**Thanks to the reviewer for the detailed comments. We respond to each point inline below.**

**Detailed review:**

Title: I think this paper is about stability more than performance, so I would suggest changing it into "A comparison of the stability and performance of ..."

**We appreciate the suggestion and have changed the title of the revised manuscript.**

Abstract: please mention that the stability analysis is performed for an explicit time discretization scheme.

**This has been added.**

page 1, line 20. Saying that the solution of Stokes problem is still infeasible is too strong of a statement. It would require a lot of resources, large clusters and parallel scalable implementations, but I would not say it's infeasible nowadays.

**This has been rephrased to "challenging".**

Eq. (6) maybe it's worth pointing out that these equations implicitly define the velocity, as the viscosity depends on the velocity. Explicit formulas can be obtained. Also, it is possible to account for a sliding term in the SIA model by adjusting the SIA velocity with the term - $\rho$  g H  $\nabla$  s /  $\beta$ . I guess this is the formulation you use in the Greenland runs.

This comment has been added, along with an explicit statement about adding a sliding term to SIA (which was not done in this work).

Section 2.2: SSA solver relies on the hypothesis that the velocity is uniform in the vertical direction. So the velocity, at any z-coordinate is the same as the depth-averaged velocity. I think it would be clearer in equations (8), (9) and (10) to substitute the velocity and the basal velocity with the averaged velocity.

**The equations have been reformulated as suggested.**

Eq. (14). I would point out that the generalized integrals depend on the velocity (except from the n=1 case).

**This clarification has been added.**

Eq. (19). One can go directly to (17) to (19) by taking the limit for \beta that goes to infinity, which gives in fact the no-slip condition.

**This was also mentioned by the second reviewer and has been changed accordingly.**

After eq. (19). The two-steps recipe to compute the velocity only holds for the case n=1, otherwise one needs to compute the vertical velocity in order to compute the viscosity needed for computing  $F_2$  in eq. (19). Please explain how the problem is solved in the general case n >= 1.

**We erroneously left out the equations for vertical shear, which can be calculated from the viscosity and basal stress from the previous iteration (see Lipscomb et al. 2019, Eq. 36). This equation has been added to the revised manuscript and this point of the algorithm clarified.**

page 10, line 1: For the stability analysis you can keep infinite domains. Just define  $x = n \Delta x$ , with n being any integer (not just natural). I think this simplifies the analysis since you do not have to worry about boundaries, and in particular you can properly invert the circulant matrix arising from the discretization. It's understood that when solving the problem in practice you'll work on finite domains and have boundary effects.

We agree that a possible conceptual route would be to start from an infinite domain. In practice, however, the issue of a finite domain cannot be avoided, and we prefer to work from this starting point. As stated in our original submission, "Our aim [in section 3] is ... to examine stability properties when applying a representative finite-difference scheme to different stress-balance equations." As such, we wish to represent the schemes as closely as possible in our analysis, as stated in the opening paragraph of this section in our original submission: "the non-local nature of the equations requires us to consider global solutions to the stress balance". Thus, considering an infinite domain leads to issues in solving the DIVA and SSA equations.

We note, however, that aside from introducing the numerical mesh, the results in Sect. 3 are independent of N, the size of the mesh, or any periodic/circulant boundary conditions. All of this is handled in the appendices, and the reader is shielded from the issues of a finite domain.

Eq. (43): Only the solution with the negative sign is acceptable.

**Our apologies, leaving both branches was an oversight.**

Section 3.2 I see a potential issue in the way the stability analysis is conducted. I would think that in most codes, when solving the thickness equation (59), the SSA and SIA velocities are discretized in the same way, whereas in the proposed analysis the SIA velocity term is expressed as a function of the thickness and treated differently. I think it would be best to consider the full velocity (usia + ussa) in (59) as a single object, take the derivative of H (usia + ussa), to get the advective and divergence terms as done in the DIVA case, and discretize them as done in the Diva case. In this way, the effective hybrid velocity should be the same as the depth-averaged hybrid velocity in (61).

Unless indicated, the equation numbers below refer to equation numbers in the new version.

We are not sure we completely understand this comment, as Eq. (59) (now 62) is simply the continuity equation using the definition of hybrid velocity, and not a discretisation.

More generally, the reviewer may be commenting on the seeming distinction with our DIVA analysis, and this apparent distinction may ultimately arise from the nature of the equations we are analysing. To carry out a stability analysis, we need an expression for depth-averaged velocity in terms of thickness (and gradients of thickness). This was stated in Sect. 3.1, but we now make it more explicit (p11, line 14):

**"To proceed, an expression for \$\boldsymbol{\delta\bar{u}}\$ in terms of \$\delta H\$ and its derivatives is needed."**

As pointed out in Sect. 3.1, this is straightforward with the SIA equations, but with DIVA and SSA, an analytical solution to the global stress balance must be solved. In the DIVA case, the solution to the DIVA stress balance then provides the full depth-averaged velocity; our solution for depth-averaged velocity can be used directly in Eq. (35), which is what we believe the reviewer means by getting "*the advective and divergence terms as done in the DIVA case*". Though it might not seem so at first, this is indeed what we have done for Hybrid and L1L2-SSA. The distinction is that with Hybrid, the depth-averaged velocity involves both the SSA contribution *and* the SIA contribution. We now make this more clear in Eq. (63, now 66) by adding two lines showing the contribution of each.

In other words: we could discretize the derivative of the (perturbed form of) H (usia + ussa), as the reviewer recommends -- but at some point we would need to replace (perturbed) ussa with the analytical solution to the SSA balance, and (perturbed) usia with its definition (cf. Eq. (63, now 66), second equality). As long as centered differences are used for all derivatives, we would end up with the same expression in Eq. (69).

**NOTE**: As a result of the question over  $u_{hyb}^{eff}$  we discovered an error in the linear perturbation in Eq. (63, now 66), which we believe is now addressed properly. The error only affects  $u_{hyb}^{eff}$ , and not the stability limits given later on. We point out that  $u_{hyb}^{eff}$ , is still distinct from that in Eq. (61, now 64), however. This is in line with other studies which show an advective velocity in the perturbed equation distinct from background velocity (e.g., Cheng et al. 2017).

Section 3.3, Similarly, I would not treat separately the terms of the L1L2 depth-averaged velocity in eq. (71), and I would avoid reformulations in (72) but simply discretize the thickness equation as for the Diva method. This might be the source of discrepancy seen when solving the problem with the CISM and Yelmo discretizations.

Please see our response above regarding major changes. We have come to agree with the reviewer that we should avoid "reformulations" – as such reformulations at the equation level are against the spirit of the analysis.

Section 3.3 is now much more brief and contains fewer equations. We formulate L1L2-SIA in the context of our simple analysis (including a term which was omitted in the original draft) and linearise the resulting scheme, but do not attempt to "reformulate away" higher-order derivatives in perturbed sliding velocity. As a result, we are unable to make further analytical headway, and so we do not take the analysis further.

We do, however, make an argument that key aspects of the discretisation can lead to more or less stability (see p17, lines 14-23); and continue this argument now in Sect. 3.4 when discussing the results, which as before show a discrepancy between the 1D model and CISM/Yelmo. A simple experiment with an alternate discretisation, discussed in Sect. 3.4 (see Eqs. (77) and (78) of the new draft), supports this argument – but it is made clear that we do not provide analytical bounds for L1L2-SIA stability.

Eq. (72) It took me a long time to figure out how the equations have been derived. I would add some more steps to make it clearer.

This comment appears to no longer be relevant due to the shortened analysis in Sect. 3.3.

Section 3.4 I think the instability in the L1L2-SIA model has been observed by BISICLES developers as well. To alleviate this, I believe BISICLES adopts a modification of the L1L2 method, consisting of avoiding the step in eq. (28), and taking the velocity to be uniform in the

vertical direction and equal to the basal velocity. It might be worth mentioning this. (In your analysis, this modified L1L2 method would be indistinguishable from the SSA method, because you consider a uniform viscosity and n=1.)

**This is now stated explicitly in the Discussion section, where it seemed to be more appropriate to put these results into a broader context.**

Section 4. Can you comment on the fact that the fitted exponent p goes from 1.5 to 2.8 instead of from 1 to 2 as in the theory. Does it depend on the Glen's law exponent n?

**The analytical solution was derived for simplified physics, with n=1 and uniform viscosity. It appears that this affects the resulting stability relationship. However, it may also be a consequence of treating more complex boundary conditions in the model on the realistic domain (e.g., a moving calving front). We have added a paragraph to discuss this explicitly in Sect. 4 as proposed.**

Page 28, Line 22. There are two possible values for r, say r+ and r-. We have that r+ > 1, which implies, see (91), that the effect of a point x on the solution at y, grows exponentially with the distance between x and y. This is clearly not physical, so r- is the only physical solution. The two solutions exist because we are considering an infinite domain. Anyway, as I mentioned before I think that there is no need to do the analysis for a finite domain. That complicates the analysis in the Appendix without really adding any value.

**Thank you for pointing this out. We have now removed the r+ solution. Please see our response to the point above ("page 10, line 1"), as to why we prefer to keep the finite domain analysis.**

**Reviewer 2 (Anonymous)**

Review of "A comparison of performance of depth-integrated ice-dynamics solvers"

Summary:

The manuscript by A. Robinson and colleagues proposes an analysis of the performance of several depth-integrated stress balance approximations by combining a theoretical analysis of the performance with a comparison of results from several existing ice flow models that include them. They start with a 1d case and then compare the results for a more realistic simulation of the Greenland ice sheet. The paper is well written and usually clear to follow, though a few steps (listed below) could use some more details to facilitate following the derivation of the stability analysis. In the case of the L1L2-SIA solver, the results are quite different between the stability analysis and the two Yelmo and CISM applications, but there is no clear explanation of this difference. I would be curious to learn what the authors think is the source of this discrepancy. Detailed comments below list relatively minor points, mostly to clarify the manuscript.

We thank the reviewer for the detailed comments. In the revised manuscript, we present results with improvements to CISM such that the results agree more closely with those of Yelmo. We also discuss further tests with the 1D model that show how changing the staggering strategy can change the mode of stability between DIVA-like and Hybrid-like curves. We believe the manuscript is now clearer on this point.

Detailed Comments:

p.1 I.6: Add that the simulations are done with finite differences and that you use von Neumann stability analysis.

Finite differences are now mentioned, also following the suggestion of Reviewer 1. We do not mention von Neumann here, as this is only done for the 1D case.

p.1 I.18: thousands -> maybe put a more exact number

This has been slightly rephrased, although the intention is to be somewhat abstract simply to illustrate the scale.

p.1 I.19: kilometers -> square kilometers

**The phrasing has been changed to refer to horizontal span in one direction.**

p.1 I.20: add that this is difficult to do at the high resolution required to accurately represent important processes

**This has been added.**

p.2 I.7: I don't think the SIA is the most widely used approximation anymore (at least not used alone), it's more historical and used to be the most common one

**We have added the qualifier "historically".**

p.4 Eq.6: I was a bit confused by this integration, it would help to put a few more details on the steps to get to this form

**This equation had an error. We have corrected the error and added an intermediate step to clarify.**

p.5 Eq.9: Is that form assumed for all the approximations? If so, mention it here.

**Yes, this clarification has been added.**

p.5 Eq.10: It is not clear how this integration is performed with the varying /mu, add indication of the main step/information needed, I could not figure out how to get to this result.

**This comment is not clear to us. In principle, there is no vertical integration needed for the SSA strain rate, as it is calculated in 2D.**

p.6 I.13: What are the implications of having a frozen bed with non zero basal stress? In which cases is that possible?

We are unsure what is meant by "implications", but we do not feel that an instance where the ice sheet is frozen to the bed and has zero basal motion is exceptional in terms of ice-sheet modelling. As discussed in Cuffey and Paterson (2010), Ch. 8, the dominant balance in glaciers and ice sheets is between basal shear stress and driving stress, and even without basal motion there can be significant basal shear stress.

p.6 Eq.19: It might be simpler to say that in that case \beta >> 1 so \beta\_eff converges toward  $1/F_2$

Yes, this is true. We have modified the logic accordingly.

p.7 l.8: "in in" -> "in its"

**This has been corrected.**

p.7 Eq.22: Add an explication about the derivation of this equation coming from the SIA and the parallel and normal components of \tau.

We added a clarifying statement, although SIA has been mentioned further below as well. We prefer not to introduce the terminology of parallel and normal, as it is not needed for this simplified description. We refer the readers to Perego et al. (2012) for more details.

p.7 I.10: maybe add that this kind of analysis is designed for finite differences

**[p.8 I.10] This clarification has been added.**

p.9 Eq.32: Add where it comes from and the assumptions made.

**We added some additional context here.**

p.9 I.33: Perturbation of what?

The ice thickness - this has been made more clear.

p.13 I.25: What is the conclusion?

This has been rephrased to indicate that the SSA solution is simply offset from the DIVA solution at coarse resolution.

p.13 I.21: Rewrite the entire equation given that assumption

The idea here is to keep the sections relating purely to SIA or SSA as short as possible, to avoid lengthening the manuscript further. If a reader wants to reproduce the SSA equation, the blueprint is given clearly, and the correct equation is already referenced.

p.17 I.9: What values are used in that range?

We added more information about the range.

p.18 I .14: I would be a bit more nuanced that it is not really the case for sliding.

The dynamic limit agrees well, but the advective limit starts to play a role for  $\Delta x < 10^3$  m. This is more clear in the revised text.

p.18 I.20: It does not really agree for  $\Delta x < 10^3$

**See the comment above.**

p.18 I.33: Why could be the reasons why it is not stable as expected with more comprehensive models?

One reason may be the model numerics related to derivative stencil choices. This is discussed in more detail in the new text.

p.19 Fig.1: Add legend for the different lines on the figure. Caption: "in some other panels" -> "In several panels"

We added some legend information and changed the caption wording as suggested.

p.20 I.8: How do you measure it?

In Yelmo, this is done by comparing the predicted ice thickness to the corrected ice thickness in a predictor-corrector scheme. This has been clarified in the text.

p.20 I.11: How much does it adaptive timestep varies temporally?

This depends on the solver being tested, as well as the domain (sometimes oscillations can be within 1 order of magnitude of the optimal timestep, sometimes oscillations are

**much smaller). It is not indicative to give a value here, but a sentence has been added to point out that oscillations do occur.**

p.26 I.1: Make sure that the results and data are actually archived

**Yes, this will be done indeed.**